# A Resilience Framework for Bi-Criteria Combinatorial Optimization with Bandit Feedback

**Vaneet Aggarwal**[*]                                             *vaneet@purdue.edu*
*Purdue University*

**Shweta Jain**                                                  *shwetajain@iitrpr.ac.in*
*IIT Ropar*

**Subham Pokhriyal**                                    *subham.22csz0002@iitrpr.ac.in*
*IIT Ropar*

**Christopher John Quinn**                                     *cjquinn@iastate.edu*
*Iowa State University*

**Reviewed on OpenReview:** *https://openreview.net/forum?id=jcjxXUMyJ5*

## Abstract

We study bi-criteria combinatorial optimization under noisy function evaluations. While resilience and black-box offline-to-online reductions have been studied in single-objective settings, extending these ideas to bi-criteria problems introduces new challenges due to the coupled degradation of approximation guarantees for objectives and constraints. We introduce a notion of $(\alpha, \beta, \delta, \mathbb{N})$-resilience for bi-criteria approximation algorithms, capturing how joint approximation guarantees degrade under bounded (possibly worst-case) oracle noise, and develop a general black-box framework that converts any resilient offline algorithm into an online algorithm for bi-criteria combinatorial multi-armed bandits with bandit feedback. The resulting online guarantees achieve sublinear regret and cumulative constraint violation of order $\tilde{O}(\delta^{2/3}\mathbb{N}^{1/3}T^{2/3})$ without requiring structural assumptions such as linearity, submodularity, or semi-bandit feedback on the noisy functions. We demonstrate the applicability of the framework by establishing resilience for several classical greedy algorithms in submodular optimization.

## 1  Introduction

Many real-world combinatorial optimization problems arising in multi-agent and learning systems require balancing competing objectives, such as minimizing cost while ensuring coverage, utility, or fairness constraints. Classical examples include sensor placement and experimental design (Krause & Guestrin, 2005; Krause et al., 2008), feature subset selection (Krause & Guestrin, 2005), and fair resource allocation (Ogryczak, 2010). These problems are naturally modeled as *bi-criteria* combinatorial optimization tasks, where one objective must be optimized subject to constraints on another.

A common approach to handling multiple objectives is to reduce them to a single objective, for example by minimizing the difference of two submodular functions (DS optimization). However, DS optimization is known to be NP-hard and inapproximable in general (Trevisan, 2014). As a result, much of the literature formulates bi-criteria problems as constrained combinatorial optimization tasks, such as minimizing a cost function subject to utility constraints. Since these problems are typically NP-hard, prior work has focused on designing approximation algorithms that achieve *bi-criteria guarantees*, relaxing both the objective and the constraint by controlled approximation factors (Iyer & Bilmes, 2012; Crawford et al., 2019; Chen et al.,

---

[*]Authors are listed in alphabetical order.

2025; Chen & Crawford, 2024). Existing analyses, however, almost universally assume access to *exact* value oracles for the objective and constraint functions (Wolsey, 1982; Wan et al., 2010; Soma & Yoshida, 2015).

In practice, exact oracle access is rarely available. Objective and constraint evaluations are often obtained through noisy measurements, simulations, or learned estimates, leading to *bounded but potentially worst-case* errors. Such perturbations can destroy structural properties such as submodularity, monotonicity, or linearity, thereby invalidating classical approximation analyses that rely on exact oracles. This difficulty is particularly acute in bi-criteria settings, where errors in the objective and constraint can interact and compound. As a result, it is unclear how to even define, let alone analyze, offline approximation guarantees that are stable enough to support online learning objectives such as regret minimization and constraint violation control under bandit feedback. This raises a fundamental and largely unresolved question:

> *How can bi-criteria approximation algorithms be made robust to bounded oracle noise in a*
> *way that enables meaningful online regret and constraint violation guarantees?*

In this work, we address this question by identifying the right offline abstraction needed to support online learning: *resilience*. While resilience and robustness notions have been studied in single-objective combinatorial optimization and bandit settings, extending them to bi-criteria problems is fundamentally nontrivial due to the coupled degradation of objective and constraint guarantees under noise. We introduce a formal notion of $(\alpha, \beta, \delta, \mathtt{N})$-*resilience* for bi-criteria approximation algorithms, which characterizes how joint approximation guarantees degrade under bounded (possibly worst-case) oracle perturbations. This notion provides an $(\alpha, \beta)$-bi-criteria guarantee with $\delta$-resilience (i.e., tolerance to perturbations in function oracles) and makes $\mathtt{N}$ oracle calls. Crucially, this definition makes no stochastic assumptions on the oracle errors and requires guarantees to hold uniformly over all oracle calls, making it strong enough to serve as a foundation for subsequent online regret and constraint violation guarantees.

Our main contribution is a *general black-box offline-to-online framework* that leverages resilience to derive online learning guarantees. We introduce a new class of problems, *Bi-Criteria Combinatorial Multi-Armed Bandits (BC-CMAB)* with bandit feedback, where the learner repeatedly selects combinatorial actions and observes only aggregate noisy reward and cost feedback. The notion of resilience captures robustness of offline bi-criteria approximation algorithms to worst-case, bounded oracle perturbations, while the online reduction operates in a stochastic bandit setting where repeated plays enable concentration. We show that any offline $(\alpha, \beta, \delta, \mathtt{N})$-resilient bi-criteria approximation algorithm can be transformed into an online algorithm achieving sublinear regret and sublinear cumulative constraint violation (CCV). The resulting guarantees scale as $\mathcal{O}(\delta^{2/3}\mathtt{N}^{1/3}T^{2/3}\log^{1/3}T)$ and require no structural assumptions such as linearity, monotonicity, submodularity, or semi-bandit feedback on the noisy functions.

As a second contribution, we establish resilience guarantees for several classical greedy algorithms in bi-criteria combinatorial optimization. In particular, we show that algorithms for Submodular Cover (SC), Submodular Cost Submodular Cover (SCSC), and Fair Submodular Maximization (FSM) satisfy $(\alpha, \beta, \delta, \mathtt{N})$-resilience under additive oracle perturbations. Proving these results requires new analysis techniques, as standard greedy proofs rely critically on exact submodularity and monotonicity. These resilience guarantees serve as concrete instantiations of the framework and immediately yield online regret and constraint violation bounds under bandit feedback. While the problems studied in the paper use submodular functions, for the framework we do not assume linearity, submodularity, or problem-specific structures on noisy functions, enabling broad applicability.

Notably, even in the single-objective setting with bandit feedback, without exploiting special reward structures such as linearity, the best known regret bounds avoiding combinatorial dependence scale as $\tilde{\mathcal{O}}(T^{2/3})$ (Nie et al., 2023). Moreover, $\Omega(T^{2/3})$ lower bounds are known for submodular maximization under bandit feedback when comparing against greedy benchmarks (Tajdini et al., 2024). Our results therefore extend the frontier of offline-to-online reductions to the bi-criteria regime, providing the first general theoretical foundation for learning with competing objectives under noisy oracle access. Unlike the single-objective case, noise simultaneously degrades both objective and constraint guarantees, and errors can propagate across criteria in a coupled manner. As a result, stability analyses based on single-objective resilience do not directly apply.

A core challenge lies in analyzing how bi-criteria guarantees degrade under noisy oracles with bounded perturbations that violate structural assumptions. For instance, the greedy analysis for SC critically relies on submodularity and exact marginal gains (Goyal et al., 2013), properties no longer preserved under noise. To address this, we develop new proof techniques: (1) We generalize density bounds (Lemma 4) for inexact oracles, ensuring greedy selections approximate the optimal sequence despite non-submodular perturbations. (2) We resolve recursive cost bounds with additive error propagation by identifying a logarithmic inequality (Lemma 5), enabling us to recover approximation ratios with graceful noise-dependent terms. (3) For termination, we prove the algorithm halts even with noisy constraints by bounding the interaction between error tolerance ($\epsilon$) and problem parameters (e.g., $\omega$, $c_{\min}/c_{\max}$), ensuring feasibility. We note that the proof of resilience is not straightforward; to illustrate this we note that where (Goyal et al., 2013) claims to prove resilience with multiplicative error, the proof has a mistake in generalizing density bounds (See Appendix C).

The key contributions of this work are as follows.

1. **Black-box offline-to-online framework.** We develop a general black-box framework that converts discrete bi-criteria offline approximation algorithms into online algorithms for *Bi-Criteria Combinatorial Multi-Armed Bandits (BC-CMAB)* with bandit feedback. The framework requires no structural assumptions, and applies to a broad class of bi-criteria combinatorial optimization problems.

2. **Resilience as the enabling offline abstraction.** We formalize a notion of $(\alpha, \beta, \delta, \mathtt{N})$-*resilience* for bi-criteria offline approximation algorithms, which characterizes how joint objective and constraint guarantees degrade under bounded (possibly worst-case) oracle noise. This notion is designed to be strong enough to support online learning guarantees while remaining agnostic to the source of noise.

3. **General online regret and constraint violation guarantees.** We show that if an offline bi-criteria approximation algorithm satisfies $(\alpha, \beta, \delta, \mathtt{N})$-resilience, then our framework produces an online BC-CMAB algorithm achieving

$$\mathcal{O}\Big(\delta^{2/3}\mathtt{N}^{1/3}T^{2/3}\log^{1/3}T\Big)$$

regret and cumulative constraint violation under bandit feedback. The guarantees depend only on the resilience parameters of the offline algorithm.

4. **Resilience of classical bi-criteria algorithms.** As concrete instantiations of the framework, we establish $(\alpha, \beta, \delta, \mathtt{N})$-resilience for several classical greedy algorithms in bi-criteria combinatorial optimization, including Submodular Cover (Goyal et al., 2013), Submodular Cost Submodular Cover (Crawford et al., 2019), and Fair Submodular Maximization (Chen et al., 2025). These results require new noise-aware analyses that go beyond existing exact-oracle proofs.

5. **Online consequences for canonical problems.** Combining the framework with the above resilience guarantees immediately yields the first sublinear regret and cumulative constraint violation guarantees for these bi-criteria problems under bandit feedback. A summary of the resulting bounds is provided in Table 1.

Overall, the proposed framework decouples online learning guarantees from problem-specific structure by isolating resilience as the key offline requirement, with classical bi-criteria optimization problems serving as representative instantiations.

## 2 Related Work

### 2.1 Offline Bi-Criteria Optimization with Combinatorial Set Selection

Offline bi-criteria optimization has been extensively studied for combinatorial problems such as submodular cover, fair submodular maximization, and knapsack-constrained optimization. Key problems include minimizing a submodular cost function while ensuring a utility threshold (Wolsey, 1982; Wan et al., 2010; Goyal et al., 2013; Crawford et al., 2019), maximizing submodular utility under fairness constraints (Chen et al.,

Table 1: Summary of the $(\alpha, \beta, \delta, \mathtt{N})$-resilient approximation for bi-criteria problems, including Submodular Cover (SC), Submodular Cost Submodular Cover (SCSC), and Fair Submodular Maximization (FSM), with the corresponding regret guarantees in CMAB under bandit feedback. This work establishes the first sublinear regret with cumulative constraint violation (CCV) under bandit feedback. Here, $\alpha$ and $\beta$ denote approximation factors for the objective $f$ and constraint $g$, respectively, where monotonicity (Mon) and submodularity (Sub) correspond to the properties of objectives and constraints. $\delta$ quantifies resilience to approximation, and $\mathtt{N}$ represents the number of oracle calls to the offline algorithm. $h \triangleq \max(f_{\max}, g_{\max})$. Details of problem-dependent parameters and other notations are discussed in Section 4.

| App. | Objective $f$ | $g$ | $\alpha$ | $\beta$ | $\delta$ | $\mathtt{N}$ | Our Regret & CCV |
|---|---|---|---|---|---|---|---|
| SC | min(Linear) | Mon+Sub | $1 + \ln\frac{\kappa}{\omega}$ | $1 - \frac{\omega}{\kappa}$ | $\frac{c_{\max}}{\omega c_{\min}} f_{\max}(3 + 6n)$ | $n^2$ | $\mathcal{O}\left(n^{4/3} f_{\max}^{5/3} T^{2/3} \log^{1/3}(T)\right)$ |
| SCSC | min(Mon+Sub) | Mon+Sub | $\rho\left(\ln\left(\frac{\Psi}{\gamma}\right) + 2\right)$ | $1$ | $\max\left\{\frac{8 c_{\max}}{c_{\min}\mu}\rho\left(\ln\left(\frac{\Psi}{\gamma}\right) + 2\right) f_{\max}, 1\right\}$ | $n^2$ | $\mathcal{O}\left(n^{4/3} h^{5/3} T^{2/3} \log^{1/3}(T)\right)$ |
| FSM | max(Mon+Sub) | Mon | $\frac{1}{1+\omega}$ | $\frac{1}{\omega}$ | $\max\left\{\frac{4\kappa}{1+\omega}, 1\right\}$ | $\frac{n\kappa}{\omega}$ | $\mathcal{O}\left(n^{1/3} f_{\max} T^{2/3} \log^{1/3}(T)\right)$ |

2025), and balancing budget adherence with objective guarantees (Iyer & Bilmes, 2013). Most works assumed exact oracles or linear rewards, limiting applicability to online settings with bandit feedback. Crawford et al. (2019) considered the offline submodular cost submodular cover problem by analyzing the approximate guarantee of a specific greedy algorithm using an inexact oracle. Goyal et al. (2013) analyzed a greedy algorithm for the submodular cover under a class of inexact oracles (multiplicative noise) for social network influence propagation. These works do not address online learning or bandit feedback and do not provide guarantees under noisy oracle perturbations (additive bounded noise).

## 2.2 CMAB with Semi-bandit Feedback

There are several works that focus on CMAB under semi-bandit feedback. Some of these include proposing a general CMAB framework under a linear reward setting (Chen et al., 2013) and an extension to a non-linear reward setting (Chen et al., 2016). A few works have also looked into Thompson sampling based algorithms to provide a general framework to solve CMAB under semi-bandit feedback (Kong et al., 2021; Wang & Chen, 2018). Adversarial CMAB with non-linear rewards is considered in (Han et al., 2021). Recent advances address constrained CMAB but remain limited to linear structures. For example, Liu et al. (2022) consider linear rewards under linear constraints; Li et al. (2023) studies best-arm identification with knapsack constraints; and Li et al. (2020) consider the linear fairness constraints in CMAB under semi-bandit feedback. Additionally, Lin et al. (2015); Yu et al. (2016); Takemori et al. (2020) analyze marginal gains as feedback under the semi-bandit setting, enabling the learner to maximize rewards with multiple constraints based on individual gains. Notably, no prior semi-bandit framework supports *bi-criteria* optimization with non-linear or combinatorial constraints.

## 2.3 Single Objective CMAB with Bandit Feedback

CMAB with bandit feedback has been widely investigated (Agarwal et al., 2021; 2022; Nie et al., 2022; 2023; 2024; Fourati et al., 2023; 2024b;a; Tajdini et al., 2024). Prior general frameworks for CMAB under bandit feedback, such as those by (Nie et al., 2023; Fourati et al., 2024a), convert offline algorithms into online algorithms using $(\alpha, \delta)$−resilience, but they focus solely on single-objective optimization. These works assume that the offline algorithm tolerates noisy reward estimates but does not address multiple objectives. Most general bandit frameworks require resilience or robustness conditions to handle noisy function estimates. Additional works include (Niazadeh et al., 2021), which presents an offline-to-online transformation for single-objective optimization under adversarial bandit feedback using the Blackwell approachability. Streeter & Golovin (2008) provide a related approach for the limited adversarial feedback in the context of single-objective submodular maximization. For instance, semi-bandit methods impose structural assumptions like monotonicity and smoothness (Chen et al., 2013), while bandit frameworks (e.g., (Nie et al., 2023)) rely on $\delta$-resilience to ensure approximation guarantees degrade gracefully with estimation errors. We note that Chen et al. (2013) considered offline "$(\alpha, \beta)$ approximation oracles," where $\alpha$ was an approximation coefficient

for the objective, but $\beta$ referred to a success probability (that the solution satisfied the $\alpha$ approximation guarantee).

Our framework addresses this gap by extending the offline algorithms to an online setting and requiring only that offline algorithms (e.g., algorithms for submodular cover (Wolsey, 1982; Goyal et al., 2013), submodular cost submodular cover (Wan et al., 2010), and fair submodular maximization (Chen et al., 2025)) satisfy $\delta$-resilience (Definition 1), a property we show holds for several existing bi-criteria approximation algorithms. This allows seamless conversion of offline guarantees to online BC-CMAB with sublinear regret and constraint violations, without problem-specific structures.

## 3 Resilience of Offline Algorithms

This section formalizes the resilience property required for offline bi-criteria approximation algorithms. Resilience ensures that small errors in evaluating the objective and constraint functions during real world use do not catastrophically degrade the performance. This property will also enable black-box conversion of offline algorithms into online bi-criteria CMAB algorithms with sublinear regret and constraint violation bounds.

**Definition 1** $((\alpha, \beta, \delta, \mathbb{N})$-Resilient Approximation). An offline algorithm $\mathcal{A}$ is an $(\alpha, \beta, \delta, \mathbb{N})$-resilient approximation algorithm for the bi-criteria problem

$$\max_{S \subseteq \Omega} f(S) \quad \text{subject to} \quad g(S) \leq \kappa,$$

if, given access to approximate deterministic oracles $\hat{f}$ and $\hat{g}$ satisfying $|f(S) - \hat{f}(S)| < \epsilon$ and $|g(S) - \hat{g}(S)| < \epsilon$ for all $S \subseteq \Omega$, $\mathcal{A}$ returns a solution $S^{\mathcal{A}}$ such that:

$$\mathbb{E}[f(S^{\mathcal{A}})] \geq \alpha f(\text{OPT}) - \delta\epsilon \quad \text{and} \quad \mathbb{E}[g(S^{\mathcal{A}})] \leq \beta\kappa + \delta\epsilon, \tag{1}$$

where $\text{OPT} = \arg\max_{S \subseteq \Omega} f(S)$ s.t. $g(S) \leq \kappa$. Here, $\mathbb{N}$ bounds the total number of oracle calls to $\hat{f}$ and $\hat{g}$, and $\delta$ quantifies resilience to approximation errors. The expectation is over the algorithms randomness; the oracle perturbations are otherwise arbitrary and need not be stochastic.

The key role of $(\alpha, \beta, \delta, \mathbb{N})$-resilient approximation is that it is exactly the condition required for the offline-to-online conversion in Section 5. Any weaker notion (e.g., multiplicative robustness or stochastic noise assumptions) would not work for such an offline-to-online conversion under bandit feedback.

When defining the resilience property on functions $f$ and $g$, one could use different parameters $\delta_f$ and $\delta_g$ where $\mathbb{E}[f(S^{\mathcal{A}})] \geq \alpha f(\text{OPT}) - \delta_f\epsilon$ and $\mathbb{E}[g(S^{\mathcal{A}})] \leq \beta\kappa + \delta_g\epsilon$. However, for the sake of simplicity, we use $\delta = \max\{\delta_f, \delta_g\}$. Also, for simple exposition, Definition 1 is defined for combinatorial bi-criteria problems (specifically over a power set).

We note that when $g$ (likewise $f$) is deterministic and known (i.e., $\hat{g}(S) = g(S)$), the resilience condition on $g$, $|g(S) - \hat{g}(S)| < \epsilon$ is not needed. Thus, we will not use/need the resilience condition on the function that is deterministic, while both inequalities in (1) will remain the same.

The $\delta$-resilience term ensures that small errors ($\epsilon$) in estimating $f$ or/and $g$ (e.g., due to noisy bandit feedback) do not compound arbitrarily. This stability is essential for real-world use, where some function values need to be approximated or estimated. It will also be important to extend offline algorithms to the online setting, where function estimates are inherently imperfect.

The definition above is for maximization problems, but can be directly extended to minimization problems $\min_{S \subseteq \Omega} f(S)$ subject to $g(S) \geq \kappa$. See Appendix A for details.

Bi-criteria algorithms under noisy feedback, such as (Crawford et al., 2019) for SCSC, demonstrate resilience to inexact oracles but remain confined to offline settings. As mentioned in (Crawford et al., 2019), if the value oracle function satisfies some nice properties such as monotonicity and submodularity, then approximation guarantees can be achieved using the existing results for SCSC. However, it is not always necessary that the value oracle function satisfies these structural properties. We extend the offline algorithm of (Crawford et al., 2019) to online BC-CMAB by formally defining the resilience guarantee and extending the approximation

---

**Algorithm 1** GREEDY-MINTSS (Goyal et al., 2013)

---

**Require:** Ground set $\Omega$, utility function $g : 2^\Omega \to \mathbb{R}^+$, cost function $f(S) = \sum_{i \in S} c_i$, threshold $\kappa$, tolerance $\omega$.

1: Initialize $S \leftarrow \emptyset$
2: **while** $g(S) < \kappa - \omega$ **do**
3:    Find $i^* \in \arg\max_{i \in \Omega \setminus S} \frac{\min(g(S \cup \{i\}), \kappa) - g(S)}{c_i}$  $\quad$ // *Maximize marginal utility per unit cost*
4:    Add $i^*$ to $S$: $S \leftarrow S \cup \{i^*\}$
5: **end while**
6: **return** $S$

---

guarantees in (Crawford et al., 2019) to resilience guarantees as a direct corollary. We further provide resilience guarantees for two more fundamental problems, SC and FSM, under any value oracle function (not necessarily monotone and submodular). We next adapt these resilience guarantees for developing an online BC-CMAB algorithm with sublinear regret guarantees.

## 4 Resilience Guarantees for Different Problems

In this section, we briefly discuss resilience of some of the bi-criteria approximation offline algorithms solving the problems of Submodular Set Cover (SC), Submodular Cost Submodular Cover (SCSC), and Fair Submodular Maximization (FSM) problems. These problems cover a wide variety of applications in social influence maximization (Goyal et al., 2013; Han et al., 2017), recommendation systems (El-Arini & Guestrin, 2011; Guillory & Bilmes, 2011), active set selection (Norouzi-Fard et al., 2016). Further, these are just some examples where we provide the resilience guarantees.

### 4.1 Resilience Guarantee for Submodular Cover Problem

The Submodular Cover (SC) problem is a minimization problem where the goal is to find a subset $S \subseteq \Omega$ that: (i) minimizes a linear non-negative cost $f(S) = \sum_{x \in S} c_x$, and (ii) satisfies $g(S) \geq \kappa$, where $g$ is a submodular utility function. Goyal et al. (2013) proposed a bi-criteria approximation algorithm GREEDY-MINTSS for this problem, which achieves the following guarantees provided access to exact oracles for $f$ and $g$. The algorithm is provided in Algorithm 1.

**Lemma 1** (Bi-Criteria Guarantees of GREEDY-MINTSS, (Goyal et al., 2013))**.** *For any $\omega > 0$, using exact oracles, GREEDY-MINTSS outputs a solution $S$ satisfying $f(S) \leq \alpha \cdot f(\text{OPT})$ and $g(S) \geq \beta \cdot \kappa$ where $\alpha = 1 + \ln(\kappa/\omega)$ and $\beta = 1 - \omega/\kappa$.*

In this work, we focus on the problem where $g$ is stochastic. There are several applications, such as online portfolio design, database query processing, sensor placement, and viral marketing, that requires online variant of submodular cover problem. For example, in the application of viral marketing the influence $g(S)$ provided by a set of nodes $S$ is generally not known beforehand and can only be obtained by sponsoring these set of influencers (nodes). Typically, the cost incurred to each sponsor $x$ is fixed and hence known beforehand, therefore we assume that $f$ is known in this setting. We show that when an inexact oracle $\hat{g}$ is used instead of $g$, with $|\hat{g}(S) - g(S)| \leq \epsilon$, GREEDY-MINTSS exhibits $\delta$-resilience. The cost guarantee degrades by an additive $\delta\epsilon$, while the utility guarantee is relaxed multiplicatively by $\beta$ and additively by $\epsilon$. Since this is a minimization problem, we use the resilience definition in Appendix A. In the following statement, we denote $c_{\max} = \max_{x \in \Omega} c_x$, $c_{\min} = \min_{x \in \Omega} c_x$, and $n = |\Omega|$.

**Theorem 1.** *For any $\omega > 0$, GREEDY-MINTSS (Goyal et al., 2013) is an $(\alpha, \beta, \delta, N)$-resilient approximation algorithm for SC with $\epsilon \leq \omega \frac{c_{\min}}{4nc_{\max}}$, where:*

$$\alpha = 1 + \ln(\kappa/\omega), \quad \beta = 1 - \omega/\kappa, \quad \delta = \frac{(3 + 6n)c_{\max}}{\omega c_{\min}} f_{\max}, \quad N = n^2.$$

See Appendix B for the proof. Since our resilience definition, Definition 1, has approximation factors $\alpha$ and $\beta$ for the objective and constraint respectively, we seek to recover the same coefficients when inexact oracles

are used. Thus, we aim to preserve the basic proof structure used in the exact oracle case (Goyal et al., 2013). Developing new variants of algorithms that are designed to be resilient to inexact oracles is a valuable research direction but out of scope for this work. We next briefly highlight some of the technical challenges encountered in the proof for inexact oracles.

We emphasize here that the existing bi-criteria proof of GREEDY-MINTSS naturally uses the submodularity and monotonicity of the exact oracle $g$. However, the inexact oracle $\hat{g}$ is neither submodular nor monotone. Every invocation of those properties in proof steps needs to be modified (if possible to do so). Another important consequence is that the sequence of sets $(S_1, S_2, \dots)$ chosen using an inexact oracle $\hat{g}$ could differ completely from the sequence chosen using an exact oracle $g$. In the following, we describe some of the key novelty in the proof.

**Generalizing Marginal Density Bounds (Lemma 4):** In the exact setting, a key component of the analysis is a marginal density bound that guarantees progress in each greedy iteration. However, this lemma assumes submodularity. In our case, since $\hat{g}$ is neither submodular nor monotone, we cannot apply this directly. More precisely, a key lemma lower-bounding marginal densities with respect to a utility gap, Lemma 4, only holds for the exact oracle $g$ and asserts a property about the density of the element $x'$ with the largest density with respect to the exact oracle $g$. (Also in (Goyal et al., 2013), the lemma was also only shown for the specific sequence of sets chosen by GREEDY-MINTSS with an exact oracle $g$.) Where that marginal density bound is used, we provide lower bound on marginal gains by the element with the largest density under the inexact oracle $\hat{g}$. We also note that for an inexact oracle, (Goyal et al., 2013) stated (without proof) a variant of their Lemma 1 for the threshold gap with respect to the inexact oracle for the unit cost (e.g., cardinality) case with multiplicative error. However, we note that this generalization is invalid, by showing a counterexample in Appendix C. Thus, we note that the procedure of establishing resilience is subtle; we show that a commonly cited generalization in (Goyal et al., 2013) does not hold under arbitrary inexact oracles.

**Logarithmic Inequality and Cost Bound Recovery:** The original proof of (Goyal et al., 2013) uses a clean recursive inequality involving exponentials to relate the cost of the final set to the optimum. However, when oracle noise is present, this recursion gains an additive error term that propagates and makes unraveling non-trivial. A major hurdle in generalizing the proof is identifying a recursive cost bound that can be used to relate the cost $f(S_{\ell-1})$ at the second to last `while` loop to the cost of the optimal solution $f(\text{OPT})$ (see proof parts 3 and 4). For the utility gap $\kappa_i$ (w.r.t. threshold $\kappa$), in the case of exact oracles in (Goyal et al., 2013), we have a recursion of the form $\kappa_i \leq \kappa_{i-1} \exp(-c_{x_i}/f(\text{OPT}))$. A simple unraveling over $\ell - 1$ iterations and using that $f$ is additive yields $\kappa_{\ell-1} \leq \kappa \exp(-f(S_{\ell-1})/f(\text{OPT}))$. Then using that $\omega < \kappa_{\ell-1}$ we can easily get the bound $f(S_{\ell-1}) \leq f(\text{OPT}) \ln(\frac{\kappa}{\omega})$.

However, working with inexact oracles we need to account for errors, and our bound has an additive error term $\kappa_i \leq \kappa_{i-1} \exp(\frac{-c_{x_i}}{f(\text{OPT})}) + \frac{3\epsilon c_{\max}}{c_{\min}}$. Unraveling yields mixed multiplicative–additive terms. Observing that the exponential term is bounded by one, we bound the terms to keep just a single additive term with $\epsilon$, $\kappa_{\ell-1} \leq \kappa \exp(-\frac{f(S_{\ell-1})}{f(\text{OPT})}) + 3\epsilon \frac{c_{\max}}{c_{\min}} \ell$.

With this formula, we can attempt to proceed like the exact oracle case and with rearranging could obtain $f(S_{\ell-1}) \leq f(\text{OPT}) \ln(\kappa/(\omega - 3\epsilon \frac{c_{\max}}{c_{\min}} \ell))$. However, since $\epsilon$ is in a logarithm with $\kappa$, when we later bound the cost of the final set $f(S_\ell)$ (see proof step 5), we would not be able to rearrange terms to recover the approximation guarantee with a coefficient $\alpha = 1 + \ln(\kappa/\omega)$ and an additive $\epsilon\delta$ term.

To recover a bound of the form $f(S_\ell) \leq (1 + \ln \frac{\kappa}{\omega}) + \epsilon\delta$, we need our bound on $f(S_{\ell-1})$ to have a similar form, $f(S_{\ell-1}) \leq \ln \frac{\kappa}{\omega} + \epsilon(\dots))$. For this we identified and proved Lemma 5 which allows us to obtain a bound on $f(S_{\ell-1})$ in that form.

**Guaranteeing Termination under Noisy Constraints:** We note that even the question of whether the algorithm reaches the stopping criteria becomes non-trivial when the criteria is based on an inexact oracle $\hat{g}$. The sequence of sets $\{S_1, S_2, \dots\}$ chosen using an inexact oracle could significantly differ from the sequence chosen using an exact oracle. If $\hat{g}(S) < g(S)$ for all sets $S$, for instance, it is possible that even though a set passed the threshold for stopping with respect to $g$, that is $g(S_i) \geq \kappa - \omega$ for some index $i$, with respect to $\hat{g}$ it has not ($\hat{g} < \kappa - \omega$) and the while loop could continue on. We show that provided $\epsilon$ is not too large with

respect to problem parameters, GREEDY-MINTSS will still reach the stopping criteria (for non-trivial thresholds $k$) even using the inexact $\hat{g}$. The range of $\epsilon \in [0, \omega \frac{c_{\min}}{4nc_{\max}}]$ is based on later analysis.

## 4.2 Resilience Guarantee for Submodular Cost Submodular Cover Problem

The Submodular Cost Submodular Cover (SCSC) problem (Wan et al., 2010) involves finding a subset $S \subseteq \Omega$ that minimizes a submodular cost function $f(S)$ while ensuring that the utility of the selected set, captured by another submodular function $g(S)$, satisfies a lower bound $\kappa$. Formally, the problem can be expressed as:

$$\text{Minimize } f(S) \quad \text{subject to} \quad g(S) \geq \kappa.$$

In this problem, both the objective $f(S)$ and the constraint $g(S)$ are submodular functions. To address this challenge, Wan et al. (2010) proposed and analyzed the GREEDY-SCSC algorithm (Algorithm 3 in Section D) assuming exact oracles. Crawford et al. (2019) analyzed GREEDY-SCSC using an inexact oracle $\hat{g}$.

**Lemma 2** (Bi-Criteria Guarantees of GREEDY-SCSC (Crawford et al., 2019)). *The GREEDY-SCSC Algorithm, when run with $\epsilon$-approximate oracle $\hat{g}$ (i.e., a fixed oracle $\hat{g}$ satisfying $|g(S) - \hat{g}(S)| \leq \epsilon$ for all $S$), returns a subset $S$ satisfying:*

$$f(S) \leq \frac{\rho}{1 - \frac{4\epsilon c_{\max}}{c_{\min}\mu}} \left( \ln\left(\frac{\Psi}{\gamma}\right) + 2 \right) f(OPT) \qquad and \qquad g(S) \geq \kappa - \epsilon,$$

*where $c_{\min} = \min_{x \in \Omega} f(\{x\})$, $c_{\max} = \max_{x \in \Omega} f(\{x\})$, $\Psi = \max_{x \in \Omega} g(\{x\})$, $\gamma = \min\{\min\{g(A_i \cup \{x\}) - g(A_i), \kappa\} : i \in [|\Omega|], x \in \Omega\}$, $\mu = \min\{g(A_i) - g(A_{i-1})\}$, where $A_i$ represents the set selected at the i-th iteration, $\rho = \max_{X \subseteq \Omega} \frac{\sum_{x \in X} f(x)}{f(X)}$ denotes the curvature of the submodular function $f$, and it is assumed that $\mu > \frac{4\epsilon c_{\max}}{c_{\min}}$.*

Lemma 2 nonlinearly combines the cost function approximation guarantee $\alpha$ (when an exact oracle is available) with the oracle error term $\epsilon$. We can decouple those terms to obtain $\delta$-resilience guarantees. To simplify the cost bound, let $\frac{4\epsilon c_{\max}}{\mu c_{\min}} \leq \frac{1}{2}$. Using the inequality $(1 - y)^{-1} \leq 1 + 2y$ for $y \leq \frac{1}{2}$, we obtain:

$$f(S) \leq \left( 1 + 2\epsilon \frac{4c_{\max}}{c_{\min}\mu} \right) \rho \left( \ln\left(\frac{\Psi}{\gamma}\right) + 2 \right) f(OPT).$$

This directly implies the following $\delta$-resilience guarantees:

**Corollary 1.** *The GREEDY-SCSC Algorithm is an $(\alpha, \beta, \delta, N)$-resilient approximation algorithm for the monotone Submodular Cost Submodular Cover problem, when $\frac{4\epsilon c_{\max}}{\mu c_{\min}} \leq \frac{1}{2}$, where:*

$$\alpha = \rho \left( \ln\left(\frac{\Psi}{\gamma}\right) + 2 \right), \quad \beta = 1,$$

$$\delta = \max\left\{ \frac{8c_{\max}}{c_{\min}\mu} \cdot \rho \left( \ln\left(\frac{\Psi}{\gamma}\right) + 2 \right) f_{\max}, 1 \right\}, \quad N = n^2.$$

## 4.3 Resilience Guarantee for Fair Submodular Maximization

Fair Submodular Maximization (FSM) is different from the previous two problems in two ways. First, it is a maximization problem of submodular function under cardinality constraints. Secondly, this problem has an additional fairness constraint, which requires that the selected set must contain the necessary fraction of elements from each group. More formally, the base set $\Omega$ is partitioned into $C$ groups represented by $\{\Omega_c\}_{c=1}^C$. The Fair Submodular Maximization problem aims to maximize a monotone submodular function $f(S)$ under cardinality and group fairness constraints. Formally,

$$\text{Maximize } f(S) \text{ s.t. } l_c \leq |S \cap \Omega_c| \leq u_c \; \forall c \in [C], |S| \leq \kappa, \tag{2}$$

where $u_c$ and $l_c$ are the upper and lower bounds for group $c$, $\kappa$ is the cardinality constraint. Chen et al. (2025) proposed a bi-criteria algorithm GREEDY-FAIRNESS-BI (see Algorithm 4 in Appendix E) for this problem. We first state their bi-criteria guarantee, with $\beta \geq 1$ relaxing the fairness constraint:

**Lemma 3** (Bi-Criteria Guarantees of GREEDY-FAIRNESS-BI, (Chen et al., 2025)). *For any $\omega \in (0,1]$ such that $1/\omega \in \mathbb{N}_+$, using exact oracles GREEDY-FAIRNESS-BI returns a subset $S$ satisfying:*

$$f(S) \geq \alpha \cdot f(OPT), where\ \alpha = \frac{1}{1+\omega},$$

$$|S \cap \Omega_c| \leq \beta u_c \quad \forall c \in [C],$$

$$\sum_{c \in C} \max\{|S \cap \Omega_c|, \beta l_c\} \leq \beta \kappa, where\ \beta = \frac{1}{\omega}$$

*Further, $\frac{n}{\omega \kappa}$ bounds the number of queries.*

**Theorem 2.** *The GREEDY-FAIRNESS-BI algorithm achieves an $(\alpha, \beta, \delta, N)$-resilient bi-criteria approximation for FSM with: $\alpha = \frac{1}{1+\omega}$, $\beta = \frac{1}{\omega}$, $\delta = \frac{4\kappa}{1+\omega}$, and $N = \frac{n\kappa}{\omega}$, where $\omega \in (0,1)$ controls the approximation-constraint trade-off, $\kappa$ is the cardinality constraint, and $n = |\Omega|$.*

The proof is provided in Appendix E.

## 5  From Resilient Offline Algorithms to Online Bandit Guarantees

We next study sequential combinatorial decision-making over a finite horizon $T$. Let $\Omega$ be a ground set of $n$ base arms and at each time step $t$, the learner selects an action $A_t \subseteq \Omega$ and observes a stochastic reward $f_t(A_t) \in [0, f_{\max}]$ and a cost $g_t(A_t) \in [0, g_{\max}]$, both drawn from unknown distributions (assumed independent across time for each fixed action) with expectations $f(A) = \mathbb{E}[f_t(A)]$ and $g(A) = \mathbb{E}[g_t(A)]$. The learners goal is to maximize the cumulative reward $\sum_{t=1}^{T} f_t(A_t)$ while ensuring that the expected cost of each action approximately satisfies a constraint $\kappa \in (0,1)$. Formally, we require: $\frac{1}{T} \sum_{t=1}^{T} g_t(A_t) \leq \kappa$.

For the offline resilience definition (Definition 1), we assume bounded oracle perturbations. In contrast, the online model studied in this section assumes stochastic bandit feedback: for each fixed action, repeated plays yield independent, bounded observations with fixed expectations, enabling concentration of empirical means. The regret and cumulative constraint violation guarantees in this section rely on this stochastic feedback model and do not extend to adversarial or adaptive bandit noise.

We note that our framework can also handle minimization problems subject to a lower bound on the utility function (see Appendix H for more details). However, for easy exposition, the framework is explained with the help of the maximization function subject to an upper bound constraint.

Since directly optimizing $f$ over a constraint on $g$ is generally NP-hard, for example, maximizing a submodular function under knapsack constraints, comparing to an exact oracle is impractical unless $T$ is exponentially large. Instead, it may be more natural to compare against what is achievable (in polynomial time) by offline approximation algorithms. Some such cases have an $(\alpha, \beta)$-bi-criteria approximation algorithm $\mathcal{A}$, where $\alpha \in (0,1]$ and $\beta \geq 1$. We define the reward regret and the cumulative constraint violation (CCV) in terms of such approximations as follows.

Let OPT denote the optimal action with respect to the expected objective and constraint functions

$$\text{OPT} \in \arg\max_{A \subseteq \Omega} f(A) \quad \text{subject to} \quad g(A) \leq \kappa.$$

The regret is defined as the gap between $\alpha$-scaled cumulative reward of the optimal feasible action and the learners reward. More formally,

$$\mathbb{E}[\mathcal{R}_f(T)] = \alpha T f(\text{OPT}) - \mathbb{E}\left[ \sum_{t=1}^{T} f_t(A_t) \right],$$

The cumulative constraint violation (CCV) measures how much the learners cumulative cost exceeds the relaxed budget $\beta T \kappa$, and is formally defined as

$$\mathbb{E}[\mathcal{V}_g(T)] = \mathbb{E}\left[\sum_{t=1}^{T} g_t(A_t)\right] - \beta T \kappa.$$

In our setting, the learner receives bandit feedback: after selecting action $A_t$, the learner observes only the reward $f_t(A_t)$ and cost $g_t(A_t)$ associated with $A_t$, with no information about other actions. We are assuming that $f_t$ and $g_t$ are stochastic—drawn from an unknown distribution with mean $f(A_t)$ and $g(A_t)$, respectively. As a special case, this also includes the cases where one of $f_t$ or $g_t$ is deterministic (i.e., $f_t(A) = f(A)$ for all $t$ or $g_t(A) = g(A)$ for all $t$). For instance, in budgeted recommendation systems, costs (e.g., monetary expenses) might be fixed and known a priori, whereas rewards (e.g., user engagement) are stochastic. However, even in such cases, the learner must still balance exploration-exploitation trade-offs for the other stochastic function. Our framework naturally accommodates both scenarios: it handles noisy $f_t$ (or $g_t$) (where $f(A_t)$ (or $g(A_t)$) is observed with randomness) and deterministic $f$ (or $g$).

## 5.1 Algorithm Description

---

**Algorithm 2** BI-CRITERIA CMAB ALGORITHM

---

**Require:** Horizon $T$, ground set $\Omega$, $(\alpha, \beta, \delta, \mathtt{N})$-resilient algorithm $\mathcal{A}$.
1: Set $m \leftarrow \left\lceil \frac{\delta^{2/3} T^{2/3} (\log T)^{1/3}}{2\mathtt{N}^{2/3}} \right\rceil$
2: **Exploration Phase:**
3: **while** $\mathcal{A}$ queries action $A$ **do**
4:   **for** $j = 1$ to $m$ **do**
5:     Play $A$, observe $f^{(j)}(A)$, $g^{(j)}(A)$
6:   **end for**
7:   Compute $\bar{f}(A) = \frac{1}{m} \sum_{j=1}^{m} f^{(j)}(A)$
8:   Compute $\bar{g}(A) = \frac{1}{m} \sum_{j=1}^{m} g^{(j)}(A)$
9:   Return $\bar{f}(A)$, $\bar{g}(A)$ to $\mathcal{A}$
10: **end while**
11: **Exploitation Phase:**
12: Let $S \leftarrow$ output of $\mathcal{A}$
13: **while** $t \leq T$ **do**
14:   Play $S$
15: **end while**

---

Our framework, BI-CRITERIA CMAB ALGORITHM (Algorithm 2), converts an offline $(\alpha, \beta, \delta, \mathtt{N})$-resilient bi-criteria approximation algorithm $\mathcal{A}$ into an online CMAB algorithm. It operates in two phases:

1. **Exploration Phase:** For each subset $A \subseteq \Omega$ queried by $\mathcal{A}$, play $A$ for $m$ rounds. In round $j$, observe noisy realizations $f^{(j)}(A)$ and $g^{(j)}(A)$ of the underlying objective and constraint functions. Define the empirical estimates

$$\bar{f}(A) \triangleq \frac{1}{m} \sum_{j=1}^{m} f^{(j)}(A), \qquad \bar{g}(A) \triangleq \frac{1}{m} \sum_{j=1}^{m} g^{(j)}(A).$$

   Return $\bar{f}(A)$ and $\bar{g}(A)$ to $\mathcal{A}$ as inexact oracle evaluations.

2. **Exploitation Phase:** Deploy $\mathcal{A}$'s output action $S$ for all remaining rounds.

## 5.2 Regret and CCV Analysis

Our framework ensures sublinear regret for the reward objective $f$ and sublinear cumulative constraint violation (CCV) for the cost constraint $g$. The theorem below formalizes these guarantees, demonstrating

that our algorithm adapts offline resilience to handle online uncertainty while balancing exploration and exploitation.

**Theorem 3** (Regret and CCV Guarantees). *For a bi-criteria CMAB instance that admits an $(\alpha, \beta, \delta, N)-$ resilient approximate offline algorithm $\mathcal{A}$, Bi-Criteria CMAB Algorithm run with $\mathcal{A}$ for a horizon $T \geq \max\left\{N, \frac{2\sqrt{2}N}{\delta}\right\}$ achieves the following $\alpha$-regret and CCV, where $h \triangleq \max(f_{\max}, g_{\max})$:*

$$\mathbb{E}[\mathcal{R}_f(T)] = \mathbb{E}[\mathcal{V}_g(T)] = \mathcal{O}\left(\delta^{2/3} h N^{1/3} T^{2/3} \log^{1/3} T\right).$$

*Remark* 1. This result represents the first bi-criteria optimization result for CMAB. Notably, it does not exploit the problem structure and avoids any combinatorial dependence on the number of arms. Additionally, Tajdini et al. (2024) established that for monotone stochastic submodular bandits with a cardinality constraint, a regret scaling of $\mathcal{O}(T^{2/3})$ is unavoidable when compared to the greedy algorithm, provided that combinatorial dependence on the arms is avoided—a necessity for small to moderate $T$.

*Proof Sketch.* We highlight a few key steps here. See Appendix G for the full proof. Let $\mathcal{E}$ denote the clean event where all empirical mean estimates satisfy $|\bar{f}(A_i) - f(A_i)| <$ rad and $|\bar{g}(A_i) - g(A_i)| <$ rad, with rad $= \sqrt{\frac{h^2 \log T}{2m}}$. Under $\mathcal{E}$, we decompose (conditional) $\alpha$-regret and CCV into separate terms for exploration and exploitation phases:

$$\mathbb{E}[\mathcal{R}_f(T)|\mathcal{E}] = \sum_{i=1}^{N} m\left(\alpha f(\text{OPT}) - \mathbb{E}[f(S_i)]\right) + (T - Nm)\left(\alpha f(\text{OPT}) - \mathbb{E}[f(S)]\right),$$

$$\mathbb{E}[\mathcal{V}_g(T)|\mathcal{E}] = \sum_{i=1}^{N} m\left(\mathbb{E}[g(S_i)] - \beta\kappa\right) + (T - Nm)\left(\mathbb{E}[g(S)] - \beta\kappa\right).$$

We bound exploration phase $\alpha$-regret and CCV using $f(\text{OPT}) \leq h$ and $g(S_i) \leq h$ respectively.

During the exploitation phase, (under $\mathcal{E}$) the $\delta$-resilience property ensures:

$$\mathbb{E}[f(S)] \geq \alpha f(\text{OPT}) - \delta \cdot \text{rad} \quad \text{and} \quad \mathbb{E}[g(S)] \leq \beta\kappa + \delta \cdot \text{rad}.$$

We can then bound the exploitation phase $\alpha$-regret and CCV respectively as:

$$(T - Nm)\left(\alpha f(\text{OPT}) - \mathbb{E}[f(S)]\right) \leq T\delta \cdot \text{rad},$$
$$(T - Nm)\left(\mathbb{E}[g(S)] - \beta\kappa\right) \leq T\delta \cdot \text{rad}.$$

We optimize $m$ as

$$m = \Theta\left(\frac{\delta^{2/3} T^{2/3} (\log T)^{1/3}}{N^{2/3}}\right)$$

to minimize total $\alpha$-regret and CCV, which leads to the result.

$\square$

*Remark* 2. Definition 1 requires the approximation guarantees to hold uniformly over all subsets $S \subseteq \Omega$, even though the online reduction in Section 5 only ever queries the offline algorithm on a finite (data-dependent) collection of sets. This stronger, uniform formulation is intentional. It allows resilience to serve as a purely offline, black-box abstraction that is independent of the internal query pattern or adaptivity of the algorithm when deployed online. In particular, the online reduction only uses the resilience guarantees on the sets actually queried, but uniform resilience ensures that these guarantees hold without requiring the online learner to reason about or restrict the offline algorithms behavior. We note that weaker notions of resilience restricted to queried sets would suffice for a fixed algorithmic instantiation, but would entangle the offline property with the online execution and reduce composability.

We note that this result can be combined with the three studied applications in the previous section. Further, the application-specific theorems impose a lower bound on the horizon $T$ through the choice of the accuracy parameter $\epsilon$, which is instantiated as the confidence radius rad in the online algorithm. The results are given as follows.

**Corollary 2.** *For the Submodular Cover problem, the Bi-Criteria CMAB Algorithm achieves the following regret and CCV bounds. For $T \geq \max\left\{n^2, \frac{2\sqrt{2}n^2\omega c_{\min}}{f_{\max}(3+6n)}\right\}$ and $\frac{T}{\log T} \geq \frac{64Nn^3 c_{\max}^3 f_{\max}^3}{\delta\omega^3 c_{\min}^3}$:*

$$\mathbb{E}[\mathcal{R}_f(T)] = \mathbb{E}[\mathcal{V}_g(T)] = \mathcal{O}\left(n^{4/3} f_{\max}^{5/3} T^{2/3} \log^{1/3}(T)\right),$$

*where $h = f_{\max} \leq nc_{\max}$.*

**Corollary 3.** *For the monotone Submodular Cost Submodular Cover problem, the Bi-Criteria CMAB Algorithm achieves the following regret and CCV bounds. For $T \geq \max\left\{N, \frac{2\sqrt{2}N}{\delta}\right\}$ and $\frac{T}{\log T} \geq \frac{512Nc_{\max}^3}{\delta\mu c_{\min}^3}$:*

$$\mathbb{E}[\mathcal{R}_f(T)] = \mathbb{E}[\mathcal{V}_g(T)] = \mathcal{O}\left(n^{4/3} h^{5/3} T^{2/3} \log^{1/3}(T)\right). \tag{3}$$

**Corollary 4.** *For the Fair Submodular Maximization problem, the Bi-Criteria CMAB Algorithm achieves the following regret and CCV bounds. For $T \geq \frac{n}{\omega}\max\{\kappa, 1+\omega\}$:*

$$\mathbb{E}[\mathcal{R}_f(T)] = \mathbb{E}[\mathcal{V}_g(T)] = \mathcal{O}\left(n^{1/3} f_{\max} T^{2/3} \log^{1/3}(T)\right).$$

## 6   Conclusions

We developed a general black-box framework for bi-criteria combinatorial optimization under noisy function evaluations. The framework is based on a notion of $(\alpha, \beta, \delta, \mathbb{N})$-resilience, which characterizes how joint objective and constraint guarantees degrade under bounded oracle perturbations, and serves as the sole offline requirement for deriving online guarantees. Using this abstraction, we showed how resilient offline approximation algorithms can be converted into online algorithms with bandit feedback, achieving sublinear regret and cumulative constraint violation without relying on structural assumptions on the noisy functions.

As instantiations, we established resilience for classical greedy algorithms in Submodular Cover, Submodular Cost Submodular Cover, and Fair Submodular Maximization, yielding the first general bi-criteria regret guarantees under bandit feedback. An important direction for future work is extending resilience and offline-to-online guarantees to broader objective classes and adversarial online feedback models.

## 7   Acknowledgement

This work is supported in part by the U.S. National Science Foundation under grants CCF-2149588 and CCF-2149617.

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

## Table of Contents

## A  Resilience Definition for Minimization

The offline algorithm $\mathcal{A}$ is an $(\alpha, \beta, \delta, \mathbb{N})$-resilient approximation for:

$$\text{Minimize } f(S) \quad \text{subject to} \quad g(S) \geq \kappa, \quad S \subseteq \Omega,$$

if, given noisy oracles $\hat{f}, \hat{g}$ with $|f(S) - \hat{f}(S)| < \epsilon$ and $|g(S) - \hat{g}(S)| < \epsilon$, it returns $S^{\mathcal{A}}$ such that:

$$\mathbb{E}[f(S^{\mathcal{A}})] \leq \alpha f(\text{OPT}) + \delta\epsilon, \tag{4}$$

$$\mathbb{E}[g(S^{\mathcal{A}})] \geq \beta\kappa - \delta\epsilon. \tag{5}$$

We also note that in this case $\alpha \geq 1$ and $\beta \leq 1$.

## B  Algorithm and Proof for the Submodular Cover Problem

**Algorithm Setup:**

- **Input**: Ground set $\Omega$, deterministic cost $f(S) = \sum_{x \in S} c_x$, utility oracle (exact $g$ or inexact $\hat{g}$), threshold $\kappa$, parameter $\omega > 0$.

- **Goal**: Minimize $f(S)$ subject to $g(S) \geq \kappa$.

- **Resilience Conditions**: For $\epsilon > 0$,

$$\mathbb{E}[f(S)] \leq \alpha f(\text{OPT}) + \delta\epsilon, \quad \mathbb{E}[g(S)] \geq \beta\kappa - \delta\epsilon.$$

The detailed offline algorithm for the problem is given in Algorithm 1, which was proposed in (Goyal et al., 2013).

We first generalize a result from (Goyal et al., 2013) that will be used in our analysis. Denote the cost function as $f(S) = \sum_{x \in S} c_x$, where $c_x$ represents the cost of the base arm $x$, which we assume is known and is not stochastic.

*Remark* 3. Lemma 1 in (Goyal et al., 2013) was shown for the specific sets chosen by the greedy algorithm using an exact value oracle ($g$). We show essentially the same proof holds for any set $S \subset \mathcal{X}$, which will be critical for our analysis when an exact value oracle is unavailable and the sequence of subsets chosen by the algorithm using $\hat{g}$ may be completely different from the sequence of subsets that would have been chosen using $g$.

**Lemma 4.** *For a non-negative, monotone non-decreasing submodular set function $g : \mathcal{X} \to \mathbb{R}^{\geq 0}$ and positive monotone cost function $f : \mathcal{X} \to \mathbb{R}_+$, for any set $S \subset \mathcal{X}$, there is an element $x \in \mathcal{X} \backslash S$ such that*

$$\frac{\min(g(S \cup \{x\}), \kappa) - \min(g(S), \kappa)}{c_x} \geq \frac{\kappa - \min(g(S), \kappa)}{f(\text{OPT})}, \tag{6}$$

*where* OPT *is the minimal cost set satisfying $g(\text{OPT}) \geq \kappa$.*

*Proof.* The proof of Lemma 4 essentially follows along the lines of Lemma 1 in (Goyal et al., 2013). As noted in (Goyal et al., 2013), thresholded monotone submodular functions, such as $\min(g(\cdot), \kappa)$ are also monotone and submodular.

If $g(S) \geq \kappa$, then the right hand side of (6) is zero. Since $g$ is monotone non-decreasing the left hand side is always non-negative, so the inequality trivially holds for any $x \in \mathcal{X} \backslash S$.

For $g(S) < \kappa$, we prove the lemma by contradiction. We will assume that for all elements $x \in \mathcal{X} \backslash S$,

$$\frac{\min(g(S \cup \{x\}), \kappa) - \min(g(S), \kappa)}{c_x} < \frac{\kappa - \min(g(S), \kappa)}{f(\text{OPT})}. \tag{7}$$

This condition means $g(S) < \kappa$. We (arbitrarily) enumerate elements in the optimal set that are not in $S$,

$$\{y_1, \ldots, y_t\} = \text{OPT}\backslash S,$$

where $t$ is the number of such elements. Since $g(S) < \kappa \le g(\text{OPT})$ we must have $t \ge 1$ (at least one element). By monotonicity, $\kappa \le g(\text{OPT}) \le g(S \cup \text{OPT})$. We have

$$
\begin{aligned}
\kappa - g(S) & \\
&= \min(g(S \cup \text{OPT}), \kappa) - \min(g(S), \kappa) && (g(\text{OPT}) \ge \kappa \text{ by def; } g(S) < \kappa \text{ by assumption}) \\
&= \sum_{i=1}^{t} \min(g(S \cup \{y_1, \ldots, y_i\}), \kappa) - \min(g(S \cup \{y_1, \ldots, y_{i-1}\}), \kappa) && (\text{telescoping sum}) \\
&\le \sum_{i=1}^{t} \min(g(S \cup \{y_i\}), \kappa) - \min(g(S), \kappa) && (\text{submodularity of } \min(g(\cdot), \kappa)) \\
&< \sum_{i=1}^{t} c_{y_i} \frac{\kappa - \min(g(S), \kappa)}{f(\text{OPT})} && (\text{using assumption Equation 7}) \\
&= f(\text{OPT}\backslash S) \frac{\kappa - \min(g(S), \kappa)}{f(\text{OPT})} && \\
&< \kappa - \min(g(S), \kappa) && (0 < f(\text{OPT}\backslash S) < f(\text{OPT})) \\
&= \kappa - g(S),
\end{aligned}
$$

a contradiction.

$\square$

We will also later use a logarithmic inequality.

**Lemma 5.** *For $a, b \in \mathbb{R}_+$ such that $\frac{b}{a} \le 0.79$, $\ln(a - b) \ge \ln(a) - \frac{2b}{a}$.*

*Proof.* First,

$$\ln(a - b) = \ln(a(1 - \frac{b}{a})) = \ln(a) + \ln(1 - \frac{b}{a}).$$

It suffices to check that $h(x) := \ln(1 - x) + 2x \ge 0$ for $0 \le x \le 0.79$. We can confirm $h(x)$ is concave with two roots.

$h'(x) = \frac{-1}{1-x} + 2$ so $h'(0) = 1$ and there is a stationary point at $x = \frac{1}{2}$.

$h''(x) = \frac{-1}{(1-x)^2}$, so $h(x)$ is concave, increasing for $x < \frac{1}{2}$ and then decreasing for $x > \frac{1}{2}$.

$h(0) = 0$ trivially. $h(0.79) \approx 0.01935225$ and $h(0.8) \approx -0.00943791$. Thus one root is $x = 0$ and the other root is in the interval $(0.79, 0.8)$.

$\square$

In the following, we will show the resilience guarantee of this algorithm, Theorem 1.

*Proof.* The proof follows along the following steps:

First, we note that even though inexact values (based on $\hat{g}(\cdot)$) are used, the algorithm will terminate for non-trivial values of the threshold $\kappa$ (i.e. $\kappa < g(\Omega)$). For any $S \subseteq \Omega$ with $g(S) \geq \kappa$ (including OPT),

$$
\begin{aligned}
\hat{g}(S) &\geq g(S) - \epsilon \\
&\geq \kappa - \epsilon \\
&\geq \kappa - \omega \frac{c_{\min}}{4nc_{\max}} \\
&\geq \kappa - \omega.
\end{aligned}
$$

**1. Noisy Utility Propagation:** The algorithm terminates when $\hat{g}(S) \geq \kappa - \omega$. Given $|\hat{g}(S) - g(S)| \leq \epsilon$,

$$
g(S) \geq \hat{g}(S) - \epsilon \geq (\kappa - \omega) - \epsilon.
$$

Rewriting for $\beta$:

$$
g(S) \geq \left(1 - \frac{\omega}{\kappa}\right) \kappa - \epsilon = \beta\kappa - \epsilon.
$$

Thus, $\beta = 1 - \frac{\omega}{\kappa}$ and the utility error term is $\delta_g \epsilon = \epsilon \implies \delta_g = 1$.

**2. Cost Error Analysis:** Let OPT $= \arg\min_{S' \subseteq \Omega}\{f(S') \mid g(S') \geq \kappa\}$. Let us denote $x_1, x_2, \ldots, x_\ell$ to be the elements added (in order) by the algorithm. Define the set $S_i = \{x_1, x_2, \ldots, x_i\}$. Thus, $S_\ell$ denotes the final set outputed by the algorithm. *We explicitly set $S_0 = \emptyset$.* We want to bound $f(S_\ell)$.

We first make two basic observations. For $i \leq \ell$, $\hat{g}(S_{i-1}) < \kappa$ since the algorithm had not yet stopped (and thus $\hat{g}(S_{i-1}) < \kappa - \omega$). We also have that $g(S_{i-1}) < \kappa$, since

$$
\begin{aligned}
g(S_{i-1}) &\leq \hat{g}(S_{i-1}) + \epsilon \\
&\leq \kappa - \omega + \epsilon \\
&\leq \kappa - \omega\left(1 - \frac{c_{\min}}{4nc_{\max}}\right) \\
&< \kappa.
\end{aligned}
$$

At each iteration, the algorithm selects $x_i$ maximizing the noisy marginal density:

$$
x_i \leftarrow \arg\max_{x \in \Omega \setminus S_{i-1}} \hat{\rho}_x(S_{i-1}) = \frac{\min(\hat{g}(S_{i-1} \cup \{x\}), \kappa) - \hat{g}(S_{i-1})}{c_x}.
$$

Let $x_i'$ denote the element with largest marginal density (with respect to the true function $g$).

$$
x_i' \leftarrow \arg\max_{x \in \Omega \setminus S_{i-1}} \rho_x(S_{i-1}) = \frac{\min(g(S_{i-1} \cup \{x\}), \kappa) - g(S_{i-1})}{c_x}.
$$

Let us further denote $\tilde{g}(S) = \min(g(S), \kappa)$. Then, by Lemma 4 (Density Bound), the largest true marginal gain satisfies

$$
\frac{\tilde{g}(S_{i-1} \cup \{x_i'\}) - \tilde{g}(S_{i-1})}{c_{x_i'}} \geq \frac{\kappa - \tilde{g}(S_{i-1})}{f(\text{OPT})}. \tag{8}
$$

We also have:

$$
\begin{aligned}
\hat{\rho}_{x_i}(S_{i-1}) &\geq \hat{\rho}_{x_i'}(S_{i-1}) && \text{(greedy selection)} \\
&\geq \rho_{x_i'}(S_{i-1}) - \frac{2\epsilon}{c_{x_i'}} && \text{(value error bound)} \\
&\geq \frac{\kappa - \tilde{g}(S_{i-1})}{f(\text{OPT})} - \frac{2\epsilon}{c_{x_i'}} && \text{(using (8))} \\
&\geq \frac{\kappa - \min(\hat{g}(S_{i-1}), \kappa)}{f(\text{OPT})} - \frac{\epsilon}{f(\text{OPT})} - \frac{2\epsilon}{c_{x_i'}} && \text{(value error bound)} \\
&\geq \frac{\kappa - \min(\hat{g}(S_{i-1}), \kappa)}{f(\text{OPT})} - 3\frac{\epsilon}{c_{\min}}.
\end{aligned}
$$

**3. Recursive Cost Bound:**

**Base case (i=1):** Let $\kappa_0 := \kappa - \min\{\hat{g}(\emptyset), \kappa\}$. Since $\hat{g}(\emptyset) \geq 0$, we have $\kappa_0 \leq \kappa$; hence the recurrence holds for $i = 1$.

Let us define the utility gap $\kappa_i = \kappa - \min(\hat{g}(S_i), \kappa)$. Then from the above inequality we get:

**General case ($i > 1$):**

$$
\begin{aligned}
\hat{\rho}_{x_i}(S_{i-1}) = \frac{\kappa_{i-1} - \kappa_i}{c_{x_i}} &\geq \frac{\kappa - \min(\hat{g}(S_{i-1}), \kappa)}{f(\text{OPT})} - 3\frac{\epsilon}{c_{\min}} \\
&= \frac{\kappa_{i-1}}{f(\text{OPT})} - 3\frac{\epsilon}{c_{\min}} \\
\implies \kappa_i &\leq \kappa_{i-1}\left(1 - \frac{c_{x_i}}{f(\text{OPT})}\right) + \frac{3\epsilon c_{\max}}{c_{\min}} \\
\implies \kappa_i &\leq \kappa_{i-1} e^{\frac{-c_{x_i}}{f(\text{OPT})}} + \frac{3\epsilon c_{\max}}{c_{\min}}
\end{aligned}
$$

**4. Telescoping Sum:**

Unrolling the recursion over $\ell - 1$ iterations, we obtain:

$$
\begin{aligned}
\kappa_{\ell-1} &\leq \kappa e^{-\sum_{i=1}^{\ell-1} \frac{c_{x_i}}{f(\text{OPT})}} + 3\epsilon \frac{c_{\max}}{c_{\min}}(\ell - 1) \\
&\leq \kappa e^{-\frac{f(S_{\ell-1})}{f(\text{OPT})}} + 3\epsilon \frac{c_{\max}}{c_{\min}}\ell.
\end{aligned} \tag{9}
$$

At termination it follows that $\kappa - \min(\hat{g}(S_\ell), \kappa) \leq \omega$ where $\omega$ is the threshold parameter. Also, because $\ell$ is the last iteration, we have $\kappa_{\ell-1} > \omega$ and $\kappa_\ell \leq \omega$.

Thus,

$$\omega \leq \kappa_{\ell-1}$$

$$\leq \kappa e^{-\frac{f(S_{\ell-1})}{f(\text{OPT})}} + 3\epsilon \frac{c_{\max}}{c_{\min}} \ell \qquad \text{(using (9))}$$

$$\implies \omega - 3\epsilon \frac{c_{\max}}{c_{\min}} \ell \leq \kappa e^{-\frac{f(S_{\ell-1})}{f(\text{OPT})}}$$

$$\implies \ln(\omega - 3\epsilon \frac{c_{\max}}{c_{\min}} \ell) \leq \ln(\kappa) - \frac{f(S_{\ell-1})}{f(\text{OPT})} \qquad (10)$$

$$\ln(\omega) - 6\epsilon \frac{c_{\max}}{\omega c_{\min}} \ell \leq \ln(\kappa) - \frac{f(S_{\ell-1})}{f(\text{OPT})} \qquad \text{(Lemma 5)}$$

$$\frac{f(S_{\ell-1})}{f(\text{OPT})} \leq \ln(\frac{\kappa}{\omega}) + 6\epsilon \frac{c_{\max}}{\omega c_{\min}} \ell$$

$$f(S_{\ell-1}) \leq f(\text{OPT}) \ln(\frac{\kappa}{\omega}) + f(\text{OPT}) 6\epsilon \frac{c_{\max}}{\omega c_{\min}} \ell, \qquad (11)$$

where for (10), since $\epsilon \leq \omega \frac{c_{\min}}{4n c_{\max}}$ and $\ell \leq n$, $\omega - 3\epsilon \frac{c_{\max}}{c_{\min}} \ell > 0$.

Since the cost function is monotone,

$$f(S_\ell) = c_{x_\ell} + f(S_{\ell-1}) \qquad (12)$$

where, $x_\ell = \arg\max_{x \in \Omega \setminus S_{\ell-1}} \hat{\rho}_x(S_{\ell-1})$. Further, let $x'_\ell = \arg\max_{x \in \Omega \setminus S_{\ell-1}} \rho_x(S_{\ell-1})$. Then,

$$\hat{\rho}_{x_\ell}(S_{\ell-1}) \geq \hat{\rho}_{x'_\ell}(S_{\ell-1})$$

$$\geq \rho_{x'_\ell}(S_{\ell-1}) - \frac{2\epsilon}{c_{x'_\ell}}$$

$$\geq \frac{\kappa - \tilde{g}(S_{\ell-1})}{f(\text{OPT})} - \frac{2\epsilon}{c_{x'_\ell}}$$

$$\geq \frac{\kappa - \min(\hat{g}(S_{\ell-1}), \kappa)}{f(\text{OPT})} - \frac{2\epsilon}{c_{x'_\ell}} - \frac{\epsilon}{f(\text{OPT})}$$

$$\geq \frac{\kappa - \min(\hat{g}(S_{\ell-1}), \kappa)}{f(\text{OPT})} - \frac{3\epsilon}{c_{\min}}$$

Thus, we get,

$$\frac{\kappa_{\ell-1} - \kappa_\ell}{c_{x_\ell}} \geq \frac{\kappa_{\ell-1}}{f(\text{OPT})} - \frac{3\epsilon}{c_{\min}}$$

Rearranging,

$$\frac{\kappa_{\ell-1} - \kappa_\ell}{\kappa_{\ell-1}} \geq \frac{c_{x_\ell}}{f(\text{OPT})} - \frac{3\epsilon}{c_{\min}} \frac{c_{x_\ell}}{\kappa_{\ell-1}}.$$

As noted above, we have $\kappa_{\ell-1} > \omega$ and $\kappa_\ell \leq \omega$. Thus $\frac{\kappa_{\ell-1} - \kappa_\ell}{\kappa_{\ell-1}} \leq 1$. Using that, we get

$$1 \geq \frac{c_{x_\ell}}{f(\text{OPT})} - \frac{3\epsilon}{c_{\min}} \frac{c_{x_\ell}}{\kappa_{\ell-1}}$$

Rearranging and using the fact that $\kappa_{\ell-1} > \omega$,

$$\frac{c_{x_\ell}}{f(\text{OPT})} \leq 1 + \frac{3\epsilon}{c_{\min}} \frac{c_{x_\ell}}{\kappa_{\ell-1}} \leq 1 + \frac{3\epsilon}{c_{\min}} \frac{c_{x_\ell}}{\omega}$$

Thus, we get

$$c_{x_\ell} \leq f(\text{OPT})\left(1 + \frac{3\epsilon}{c_{\min}}\frac{c_{\max}}{\omega}\right). \tag{13}$$

**5. Resilience Parameter $\delta$ and Oracle Calls N:**

Combining (11), (12), and (13),

$$
\begin{aligned}
f(S_\ell) = c_{x_\ell} + f(S_{\ell-1}) \\
\leq f(\text{OPT})\left(1 + \frac{3\epsilon}{c_{\min}}\frac{c_{\max}}{\omega}\right) + f(\text{OPT})\ln(\frac{\kappa}{\omega}) + f(\text{OPT})6\epsilon\frac{c_{\max}}{\omega c_{\min}}\ell \\
= f(\text{OPT})\left(1 + \ln\left(\frac{\kappa}{\omega}\right)\right) + \epsilon\frac{c_{\max}}{\omega c_{\min}}f(\text{OPT})(3 + 6\ell) \qquad \text{(rearranging)} \\
\leq f(\text{OPT})\left(1 + \ln\left(\frac{\kappa}{\omega}\right)\right) + \epsilon\frac{c_{\max}}{\omega c_{\min}}f_{\max}(3 + 6n).
\end{aligned}
$$

Thus, we get $\alpha = 1 + \ln\left(\frac{\kappa}{\omega}\right)$ and $\delta = \frac{c_{\max}}{\omega c_{\min}}f_{\max}(3 + 6n)$.

Each iteration selects one element and there are at most $n$ elements. Further, each iteration queries $g$ at most $n$ times. Thus, $\text{N} = \frac{1}{2}n(n + 1)$ though for simplicity we use $\text{N} = n^2$. $\qquad\square$

## C   Counter-example to (Goyal et al., 2013)'s generalization of their Lemma 1 for (multiplicative) inexact oracles for GREEDY-MINTSS

We note that the authors of (Goyal et al., 2013) claimed that it is 'straightforward' to generalize their result to (multiplicative) inexact oracles, while we note in the following that such generalization does not hold. We point this out to highlight that care must be taken in extending results involving exact oracles to inexact oracles.

Consider a modular (thus also monotone submodular) function constraint function $f$ (they use $f$ where we write $g$) with $f(1) = 0.99$, $f(2) = 0.01$, $f(i) = 0$ for $i = 3, 4, \ldots$. Using (Goyal et al., 2013)'s notation, for the threshold $\eta = 1$, clearly for the unit-cost case (the only case explicitly considered in (Goyal et al., 2013) for inexact oracles), the optimal set is $1, 2$ and optimal cost is cardinality $k = 2$.

Consider a inexact oracle $f'$, with $f'(1) = 0.02$, $f'(2) = 0.01$, with other values equal to 0. The greedy algorithm using $f'$ would pick element 1 and then 2, which is also the sequence the greedy algorithm would pick using the exact oracle. (Goyal et al., 2013)'s claimed generalization of their Lemma 1 for this case after the first step has that there must exist an element $x \in \Omega\backslash 1$ satisfying

$$f(1, x) - f(1) \geq (\eta - f'(1))/k = (1 - 0.02)/2 = 0.98/2 = 0.47,$$

which is not true. They use this claim directly in their proof of resilience for multiplicative noise, and thus their resilience proof as written is not correct.

## D   GREEDY-SCSC Algorithm

Here we include the pseudo-code for GREEDY-SCSC by (Wan et al., 2010).

---
**Algorithm 3** Greedy-SCSC
---
1: **Require:** Submodular oracle $g$, submodular cost function $f$ and threshold $\kappa$.
2: Initialize $S \leftarrow \emptyset$.
3: **while** $g(S) < \kappa$ **do**
4: $\quad u \leftarrow \underset{i \in \Omega \setminus S}{\arg\max} \frac{\min(g(S \cup \{i\}), \kappa) - \min(g(S), \kappa)}{f(\{i\})}$.
5: $\quad$ Update $S \leftarrow S \cup \{u\}$.
6: **end while**
7: **return** $S$.
---

# E   Algorithm and Proof of Fair Submodular Maximization

---
**Algorithm 4** Greedy-fairness-bi
---
1: **Require:** $\quad S \leftarrow \emptyset$, $\omega$, Partition set $P = \{\Omega_1, \Omega_2, \ldots, \Omega_C\}$, approximate fairness matroid $\mathcal{M}_{1/\omega} = \mathcal{M}_{1/\omega}(P, \kappa/\omega, \vec{l}/\omega, \vec{u}/\omega)$
2: **Output:** $S \in \Omega$
3: **while** $\exists i$ s.t. $S \cup \{i\} \in \mathcal{M}_{1/\omega}$ **do**
4: $\quad V \leftarrow \{i \in \Omega \mid S \cup \{i\} \in \mathcal{M}_{1/\omega}\}$
5: $\quad$ Find $i^* \in \arg\max_{i \in V \setminus S} (f(S \cup \{i\}) - f(S))$
6: $\quad$ Add $i^*$ to $S$: $S \leftarrow S \cup \{i^*\}$
7: **end while**
8: **return** $S$
---

Before proving Theorem 2, we first state a technical lemma from (Chen et al., 2025) relating the feasible regions of the problems with strict ($1/\omega = 1$) and relaxed ($1/\omega > 1$) fairness constraints, characterized as matroids $\mathcal{M}_{1/\omega}(P, \kappa/\omega, \vec{l}/\omega, \vec{u}/\omega) = \{S \subseteq \Omega : |S \cap \Omega_c| \leq \frac{u_c}{\omega}, \forall c \in [C], \sum_{c \in [C]} \max\{|S \cap \Omega_c|, \frac{l_c}{\omega}\} \leq \frac{\kappa}{\omega}\}$.

**Lemma 6** ((Chen et al., 2025)). *For any set $S \in \mathcal{M}_{1/\omega}(P, \kappa/\omega, \vec{l}/\omega, \vec{u}/\omega)$ with $|S| = \frac{\kappa}{\omega}$, $T \in \mathcal{M}_1(P, \kappa, \vec{l}, \vec{u})$, with $|T| = \kappa$, and any permutation of $S = (s_1, s_2, \ldots, s_{\kappa/\omega})$, there exists a sequence $E = (e_1, e_2, \ldots e_{\kappa/\omega})$ such that each element in $T$ appears $1/\omega$ times in $E$ and that $S_i \cup \{e_{i+1}\} \in \mathcal{M}_{1/\omega}, \quad \forall i \in \{0, 1, \ldots, \kappa/\omega\}$ where $S_i = (s_1, s_2, \ldots, s_i)$ and $S_0 = \emptyset$.*

*Proof of Theorem 2.* We begin by denoting the optimal solution of the problem as OPT $= \arg\max_{S \in \mathcal{M}_1(P, \kappa, \vec{l}, \vec{u})} f(S)$. For iteration $i = 1, \ldots, \kappa/\omega$, let $S_i$ denote the set selected in that iteration. Apply Lemma 6 with OPT as $T$ to obtain a valid sequence $E$ of $1/\omega$ copies of OPT.

Since Algorithm 4 stops after $\kappa/\omega$ steps and outputs a set $|S| = \kappa/\omega$ and $S \in \mathcal{M}_{1/\omega}$, we have $|S \cup \Omega_c| \leq \frac{u_c}{\omega} \quad \forall c \in [C]$ and $\sum_{c \in [C]} \max\{|S \cap \Omega_c|, l_c/\omega\} \leq \kappa/\omega$. These two equations directly give resilience on cardinality as $\beta = 1/\omega$ and $\delta_g = 0$.

Consider the set $S_{i+1}$ chosen in the $(i+1)$th iteration. Since the algorithm chose the element $S_{i+1} \setminus S_i$ instead of $e_{i+1} \in$ OPT such that $S_i \cup \{e_{i+1}\} \in \mathcal{M}_{1/\omega}$,

$$
\begin{aligned}
f(S_{i+1}) - f(S_i) &\geq \hat{f}(S_{i+1}) - \hat{f}(S_i) - 2\epsilon \\
&\geq \hat{f}(S_i \cup \{e_{i+1}\}) - \hat{f}(S_i) - 2\epsilon \\
&\geq f(S_i \cup \{e_{i+1}\}) - f(S_i) - 4\epsilon \\
&\geq f(S \cup \{e_{i+1}\}) - f(S) - 4\epsilon,
\end{aligned}
$$

where the last step uses that $f$ is submodular. Sum both sides of the last inequality over all iterations $i = 0, 1, \ldots, \kappa/\omega - 1$. Now, $\sum_{i=0}^{\kappa/\omega-1} f(S_{i+1}) - f(S_i) = f(S) - f(\emptyset) = f(S)$. Also, from Lemma 6, each $e_{i+1} \in E$, where $E$ is a sequence containing $1/\omega$ copies of each element in OPT. Therefore, $\sum_{i=0}^{\kappa/\omega-1} f(S \cup \{e_{i+1}\}) - f(S)$

is equal to $1/\omega \sum_{i^* \in OPT} f(S \cup i^*) - f(S)$. Using a well-known identity for monotone submodular functions,

$$\sum_{i^* \in \text{OPT}} f(S \cup \{i^*\}) - f(S) \geq f(\text{OPT}) - f(S).$$

Consequently, we get:

$$f(S) \geq 1/\omega \left[ \sum_{i^* \in \text{OPT}} f(S \cup \{i^*\}) - f(S) \right] - 4\epsilon \frac{\kappa}{\omega} \geq \frac{f(\text{OPT}) - f(S)}{\omega} - 4\epsilon \frac{\kappa}{\omega}.$$

Rearranging terms and observing that since the algorithm runs for $\kappa/\omega$ steps, and uses at most $n$ oracle calls in each step, the total oracle calls are bounded by $\mathbb{N} = \frac{n\kappa}{\omega}$. $\qquad \square$

## F   Clean Event Bound

We define key events for our analysis. For each action $A$ played during exploration, the $m$ observed rewards are i.i.d. with mean $f(A)$ and are bounded in $[0, f_{\max}]$. Likewise, the $m$ observed constraint values are i.i.d. with mean $g(A)$ and are bounded in $[0, g_{\max}]$. For simplicity, we will use the bound $h = \max\{f_{\max}, g_{\max}\}$ By Hoeffdings inequality, the empirical means $\bar{f}(A)$ and $\bar{g}(A)$ satisfy the respective concentration bounds

$$\mathbb{P}\left(\left|\bar{f}(A) - f(A)\right| \geq \epsilon\right) \leq 2\exp\left(-\frac{2m\epsilon^2}{h^2}\right) \quad \text{and} \quad \mathbb{P}\left(\left|\bar{g}(A) - g(A)\right| \geq \epsilon\right) \leq 2\exp\left(-\frac{2m\epsilon^2}{h^2}\right).$$

These bounds hold for all actions played during exploration.

**Lemma 7** (Concentration of Empirical Means in Exploration). *Let $A_1, \ldots, A_N$ be the set of actions played during the exploration phase, each played $m$ times. Suppose the rewards and constraints associated with these actions are bounded in $[0, h]$, and let $\bar{f}(A_i)$ and $\bar{g}(A_i)$ denote of action $A_i$'s empirical mean reward and the empirical mean constraint value respectively. Define the confidence radius*

$$\text{rad} := \sqrt{\frac{h^2 \log T}{2m}}.$$

*Then, with probability at least $1 - 4NT^{-1}$, the normalized empirical means of all actions remain within* rad *of their true means, i.e.,*

$$\mathcal{E} := \bigcap_{i=1}^{N} \left\{\left|\bar{f}(A_i) - f(A_i)\right| < \text{rad}\right\} \cap \left\{\left|\bar{g}(A_i) - g(A_i)\right| < \text{rad}\right\}$$

*holds.*

*Proof.* Applying Hoeffdings inequality to each action $A_i$, we obtain

$$\mathbb{P}\left(\left|\bar{f}(A_i) - f(A_i)\right| \geq \text{rad}\right) \leq 2\exp\left(-\frac{2m\text{rad}^2}{h^2}\right)$$

and

$$\mathbb{P}\left(\left|\bar{g}(A_i) - g(A_i)\right| \geq \text{rad}\right) \leq 2\exp\left(-\frac{2m\text{rad}^2}{h^2}\right).$$

Substituting $\text{rad} = \sqrt{h^2 \log(T)/2m}$, we compute

$$\mathbb{P}\left(\left|\bar{f}(A_i) - f(A_i)\right| \geq \text{rad}\right) \leq 2T^{-1} \quad \text{and} \quad \mathbb{P}\left(\left|\bar{g}(A_i) - g(A_i)\right| \geq \text{rad}\right) \leq 2T^{-1}.$$

Define the event that the empirical means for both the objective and constraint functions are within the confidence bound for the $i$th action as $\mathcal{E}_i = \{|\bar{f}(A_i) - f(A_i)| < \text{rad}\} \cap \{|\bar{g}(A_i) - g(A_i)| < \text{rad}\}$.

Denote the complement of $\mathcal{E}_i$ as

$$\mathcal{E}_i^c = \{|\bar{f}(A_i) - f(A_i)| \geq \text{rad}\} \cup \{|\bar{g}(A_i) - g(A_i)| \geq \text{rad}\}.$$

By the union bound,

$$\mathbb{P}\left(\mathcal{E}_i^c\right) \leq \mathbb{P}\left(\left|\bar{f}(A_i) - f(A_i)\right| \geq \text{rad}\right) + \mathbb{P}\left(\left|\bar{g}(A_i) - g(A_i)\right| \geq \text{rad}\right)$$
$$\leq 4T^{-1}.$$

Let $\mathcal{E}^c$ denote the complement of the clean event $\mathcal{E}$. Using the union bound,

$$\mathbb{P}\left(\mathcal{E}^c\right) = \mathbb{P}\left(\bigcup_{i=1}^{N} \mathcal{E}_i^c\right) \leq \sum_{i=1}^{N} \mathbb{P}\left(\mathcal{E}_i^c\right) \leq \sum_{i=1}^{N} 4T^{-1} = 4NT^{-1}.$$

Thus,

$$\mathbb{P}(\mathcal{E}) = 1 - \mathbb{P}(\mathcal{E}^c) \geq 1 - 4NT^{-1},$$

completing the proof.

## G  Proof of Theorem 3

*Proof.* Let $\mathcal{E}$ denote the clean event where all empirical mean estimates satisfy $|\bar{f}(A_i) - f(A_i)| < \text{rad}$ and $|\bar{g}(A_i) - g(A_i)| < \text{rad}$, with $\text{rad} = \sqrt{\frac{h^2 \log T}{2m}}$.

From Lemma 7, $\mathbb{P}(\mathcal{E}) \geq 1 - 4NT^{-1}$.

Using the law of total expectation, decompose the $\alpha$-regret and CCV:

$$\mathbb{E}[\mathcal{R}_f(T)] = \mathbb{E}[\mathcal{R}_f(T)|\mathcal{E}]\mathbb{P}(\mathcal{E}) + \mathbb{E}[\mathcal{R}_f(T)|\mathcal{E}^c]\mathbb{P}(\mathcal{E}^c), \tag{14}$$

$$\mathbb{E}[\mathcal{V}_g(T)] = \mathbb{E}[\mathcal{V}_g(T)|\mathcal{E}]\mathbb{P}(\mathcal{E}) + \mathbb{E}[\mathcal{V}_g(T)|\mathcal{E}^c]\mathbb{P}(\mathcal{E}^c). \tag{15}$$

Under $\mathcal{E}$, further decompose both quantities into separate terms for exploration and exploitation phases:

$$\mathbb{E}[\mathcal{R}_f(T)|\mathcal{E}] = \sum_{i=1}^{N} m\left(\alpha f(\text{OPT}) - \mathbb{E}[f(S_i)]\right)$$
$$+ (T - Nm)\left(\alpha f(\text{OPT}) - \mathbb{E}[f(S)]\right), \tag{16}$$

$$\mathbb{E}[\mathcal{V}_g(T)|\mathcal{E}] = \sum_{i=1}^{N} m\left(\mathbb{E}[g(S_i)] - \beta\kappa\right) + (T - Nm)\left(\mathbb{E}[g(S)] - \beta\kappa\right). \tag{17}$$

During the exploration phase, we have the following. Since $f(S_i) \leq h$,

$$\sum_{i=1}^{N} m\left(\alpha f(\text{OPT}) - \mathbb{E}[f(S_i)]\right) \leq \alpha Nmh. \tag{18}$$

Since $g(S_i) \leq h$,

$$\sum_{i=1}^{N} m\left(\mathbb{E}[g(S_i)] - \beta\kappa\right) \leq Nmh. \tag{19}$$

During the exploitation phase, we have the following. Under $\mathcal{E}$, the $\delta$-resilience property ensures:

$$\mathbb{E}[f(S)] \geq \alpha f(\text{OPT}) - \delta \cdot \text{rad}, \quad \mathbb{E}[g(S)] \leq \beta\kappa + \delta \cdot \text{rad}. \tag{20}$$

Substituting into (16) and (17):

$$(T - Nm)\left(\alpha f(\text{OPT}) - \mathbb{E}[f(S)]\right) \leq T\delta \cdot \text{rad}, \tag{21}$$

$$(T - Nm)\left(\mathbb{E}[g(S)] - \beta\kappa\right) \leq T\delta \cdot \text{rad}. \tag{22}$$

In order to equate exploration and exploitation terms for $m$ in order, we choose

$$m = \Theta\left(\frac{\delta^{2/3}T^{2/3}(\log T)^{1/3}}{N^{2/3}}\right). \tag{23}$$

Substituting (23) into (18), (19), (21), and (22):

$$\mathbb{E}[\mathcal{R}_f(T)|\mathcal{E}] = \mathcal{O}\left(\delta^{2/3}hN^{1/3}T^{2/3}\log^{1/3}T\right), \tag{24}$$

$$\mathbb{E}[\mathcal{V}_g(T)|\mathcal{E}] = \mathcal{O}\left(\delta^{2/3}hN^{1/3}T^{2/3}\log^{1/3}T\right). \tag{25}$$

The bad event contribution can be bounded as follows. For $\mathcal{E}^c$ with $\mathbb{P}(\mathcal{E}^c) \leq 4NT^{-1}$:

$$\mathbb{E}[\mathcal{R}_f(T)|\mathcal{E}^c]\mathbb{P}(\mathcal{E}^c) \leq 4Nh = \mathcal{O}(1), \tag{26}$$

$$\mathbb{E}[\mathcal{V}_g(T)|\mathcal{E}^c]\mathbb{P}(\mathcal{E}^c) \leq 4Nh = \mathcal{O}(1). \tag{27}$$

From (14)–(15), (24)–(25), and (26)–(27):

$$\mathbb{E}[\mathcal{R}_f(T)] = \mathcal{O}\left(\delta^{2/3}hN^{1/3}T^{2/3}\log^{1/3}T\right),$$

$$\mathbb{E}[\mathcal{V}_g(T)] = \mathcal{O}\left(\delta^{2/3}hN^{1/3}T^{2/3}\log^{1/3}T\right).$$

$\square$

$\square$

## H    The Framework Applied to Minimization Problem

This appendix details the conversion of the bi-criteria CMAB framework from a maximization problem (Section 5) to a minimization problem. We redefine the problem setup and regret guarantees for the minimization setting.

### H.1    Problem Statement for Minimization

The learners goal is to minimize a cumulative *cost* $\sum_{t=1}^{T} f_t(A_t)$ while ensuring that the expected *utility* of each action approximately satisfies a lower-bound constraint $\kappa \in \mathbb{R}^+$. Formally, we require:

$$\frac{1}{T}\sum_{t=1}^{T} g_t(A_t) \geq \kappa.$$

Let OPT denote the optimal action for the minimization problem:

$$\text{OPT} \in \arg\min_{A \subseteq \Omega} f(A) \quad \text{subject to} \quad g(A) \geq \kappa.$$

The regret and cumulative constraint violation (CCV) are redefined as:

$$\mathbb{E}[\mathcal{R}_f(T)] = \mathbb{E}\left[\sum_{t=1}^{T} f_t(A_t)\right] - \alpha T f(\text{OPT}),$$

$$\mathbb{E}[\mathcal{V}_g(T)] = \beta T \kappa - \mathbb{E}\left[\sum_{t=1}^{T} g_t(A_t)\right],$$

where $\alpha \geq 1$ is the cost approximation factor and $\beta \leq 1$ is the utility relaxation factor.

## H.2 Modified Framework and Analysis

The online algorithm (Algorithm 2) remains unchanged, but the analysis adapts to the minimization objective:

**Theorem 4** (Regret and CCV for Minimization). *For any bi-criteria CMAB minimization instance with horizon $T \geq \max\left\{N, \frac{2\sqrt{2}N}{\delta}\right\}$ and $h \triangleq \max(f_{\max}, g_{\max})$, BI-CRITERIA CMAB ALGORITHM achieves:*

1. *Expected $\alpha$-regret:*

$$\mathbb{E}[\mathcal{R}_f(T)] = \mathcal{O}\left(\delta^{2/3} h N^{1/3} T^{2/3} \log^{1/3} T\right),$$

2. *Expected cumulative $\beta$-constraint violation:*

$$\mathbb{E}[\mathcal{V}_g(T)] = \mathcal{O}\left(\delta^{2/3} h N^{1/3} T^{2/3} \log^{1/3} T\right).$$

*Proof Sketch.* The proof follows the same structure as Theorem 3, with adjustments for minimization:

1. **Clean Event**: Concentration bounds hold as in Lemma 7.

2. **Resilience Guarantees**: Under clean event $\mathcal{E}$, from Appendix A, we have

$$\mathbb{E}[f(S)] \leq \alpha f(\text{OPT}) + \delta h \text{rad},$$
$$\mathbb{E}[g(S)] \geq \beta \kappa - \delta h \text{rad}.$$

3. **Regret and CCV Decomposition**:

$$\mathbb{E}[\mathcal{R}_f(T)|\mathcal{E}] = \sum_{i=1}^{N} m(\mathbb{E}[f(S_i)] - \alpha f(\text{OPT})) + \sum_{t=Nm+1}^{T} (\mathbb{E}[f(S)] - \alpha f(\text{OPT})),$$

$$\mathbb{E}[\mathcal{V}_g(T)|\mathcal{E}] = \sum_{i=1}^{N} m(\beta \kappa - \mathbb{E}[g(S_i)]) + \sum_{t=Nm+1}^{T} (\beta \kappa - \mathbb{E}[g(S)]).$$

4. **Bounding Terms**: Exploration regret $\leq N m f_{\max}$, exploitation regret $\leq T \delta h \text{rad}$. Similar bounds apply to CCV, where exploration CCV is $\leq N m \beta \kappa$ and exploitation CCV $\leq T \delta h \text{rad}$.

5. **Hyperparameter Substitution**: Substituting $m = \mathcal{O}\left(\frac{\delta^{2/3} T^{2/3} \log^{1/3} T}{N^{2/3}}\right)$ balances exploration and exploitation terms.

The full proof mirrors Theorem 3, with inequalities modified as above to reflect minimization. $\square$

## H.3 Key Differences from Maximization Framework

1. **Objective and Constraint Swap**: Cost minimization replaces reward maximization; utility lower-bound replaces cost upper-bound.

2. **Regret/CCV Definitions**: Regret measures excess cost, while CCV measures utility shortfall.

3. **Resilience Inequalities**: Additive errors $\delta\epsilon$ increase cost bounds and decrease utility bounds.

This conversion demonstrates the frameworks flexibility in handling dual objectives across maximization and minimization problems under bandit feedback.

