# OpenReview forum: "A Resilience Framework for Bi-Criteria Combinatorial Optimization with Bandit Feedback"
_TMLR — Accepted by TMLR_

### Review · Reviewer_dK8f · 2026-01-27

**Summary Of Contributions:**

This submission studies bi-criteria combinatorial optimization when function evaluations are noisy, and connects offline bi-criteria approximation to online learning with bandit feedback via a new stability notion (“resilience”).

- **A bi-criteria resilience definition.** The paper introduces **(α, β, δ, N)-resilience** for bi-criteria approximation algorithms, intended to quantify how _both_ the objective approximation and constraint satisfaction degrade under bounded (potentially adversarial) oracle perturbations, while tracking the number of oracle calls **N** and the noise-sensitivity **δ**.

- **A black-box offline-to-online reduction for bi-criteria CMAB with bandit feedback.** Given an offline algorithm with such resilience, the paper provides a generic **explore-then-exploit** conversion that estimates queried action values via repeated bandit plays and feeds empirical means as approximate oracles to the offline routine.

- **General regret and cumulative constraint violation bounds.** The resulting online algorithm achieves **sublinear α-regret** and **sublinear cumulative constraint violation (CCV)** with rate on the order of

    $\tilde{O}\left(\delta^{2/3}hN^{1/3}T^{2/3}\right)$

    (up to log factors), without assuming linearity/submodularity/semi-bandit feedback on the _noisy_ functions.

- **Resilience proofs for several canonical greedy bi-criteria algorithms.** The paper establishes resilience (with explicit parameters) for greedy algorithms in **Submodular Cover (SC)**, **Submodular Cost Submodular Cover (SCSC)**, and **Fair Submodular Maximization (FSM)**, including handling the technical issue that noise can destroy submodularity/monotonicity of the observed oracle.

- **A diagnostic/correction regarding prior resilience claims.** It provides a counterexample to a claimed extension in prior work (as stated in the paper), highlighting subtlety in generalizing greedy density arguments under inexact oracles.



Key strengths:

- **Clean abstraction** (bi-criteria resilience) that is well-matched to an offline-to-online reduction.
- **General reduction** that does not rely on semi-bandit feedback or special reward structure (beyond boundedness + concentration).
- **Nontrivial technical work** in proving resilience for greedy methods despite the loss of submodularity/monotonicity under perturbed oracles.
- **Clear theorem statements** that expose how online learning performance depends on offline algorithmic properties (δ and N).


Key weaknesses:

- The reduction is essentially **explore-then-commit**, yielding **T^{2/3}** rates; this is consistent with known barriers in structure-free bandit feedback, but it limits practicality when better rates are possible under additional structure.
- The resilience definition appears **strong/uniform** (it is stated for all subsets), while the online reduction only needs accuracy on the **queried** subsets; this mismatch could confuse readers and should be reconciled more explicitly.
- The algorithm requires (or implicitly uses) knowledge/upper bounds of **δ, N, h** (and horizon **T**) to set exploration length; it is unclear how robust the method is to misspecification or how to tune in practice.
- The paper is heavily theoretical and (as provided) does not include empirical validation; while not required, experiments could materially improve confidence and interpretability of constants and regimes.

**Audience:**

Yes

**Audience Explanation:**

- The work sits at the intersection of **approximation algorithms**, **online learning/bandits**, and **constrained optimization**, all of which are well represented in TMLR’s readership.

- The proposed abstraction (bi-criteria resilience) is a **conceptual contribution** that may be reusable beyond the three worked examples—especially for researchers exploring stability of combinatorial optimization under noisy evaluation and those building oracle-based online learning algorithms.

- The reduction from offline algorithm to online algorithms in the setting of **bandit feedback** for combinatorial actions with competing objectives/constraints is a meaningful attempt that could arouse interests in deepening the techniques and the corresponding proof methodologies.

**Broader Impact Concerns:**

The submission is primarily methodological/theoretical, no major ethical implications or concerns that would require such statement.

**Claims And Evidence:**

Yes

**Claims Explanation:**

The central claims are theoretical (definition + reduction + regret/CCV bounds + resilience proofs).

A formal resilience definition for bi-criteria approximation algorithms and an explanation of why it is the “right” offline property for their online conversion are given.

Problem-specific resilience theorems/corollaries for SC, SCSC, and FSM, including explicit parameter dependencies are provided (proofs in appendix, can be verified with coherency).

The authors have also provided an explicit offline-to-online algorithm description (two-phase exploration/exploitation) together with a proof sketch/derivation of the regret and CCV bounds using concentration (Hoeffding) and the offline resilience guarantee.


Parameter assumptions and techniques to overcome major hurdles when compared to classical proofs are stated. However, assumptions in the online model should be stated more crisply. For e.g., the analysis relies on i.i.d. sampling/concentration for each queried action and bounded rewards/costs; the text suggests stochastic draws with fixed means, but the exact independence/stationarity assumptions (and whether reward and cost are independent) are not explicitly. While not completely necessary as these assumptions are well-known among domain experts, they do increase the rigor the of a theory paper.

**Requested Changes:**

1. **Clarify the stochastic model and independence assumptions used in Theorem 3 (and related results).**
    Concretely: specify whether, for each action $(A)$, (${f_t(A)}_t$) and (${g_t(A)}_t$) are i.i.d. across time with fixed means; whether reward and cost are independent conditional on $(A)$; and whether any adaptive dependence is allowed. This is essential because the proof uses concentration for empirical means.
2. **Reconcile the resilience definition (uniform oracle error for all sets) with what the online reduction actually requires (accuracy only on queried sets).**

---

> ### Author Response · Authors · 2026-02-19
>
> Thank you for your feedback.  Below we reply to several comments, followed by discussions of the requested changes.
>
>
> > The reduction is essentially explore-then-commit, yielding $T^{2/3}$ rates; this is consistent with known barriers in structure-free bandit feedback, but it limits practicality when better rates are possible under additional structure.
>
> **Response.**
> Our intent is not to claim optimality even under additional structure (i.e., for linear MAB, $\sqrt{T}$ regret methods are known)  but rather to establish the first general offline-to-online reduction for bi-criteria problems under bandit feedback, and cleanly separate what comes from offline algorithmic stability (via $\delta$, $N$) from what comes from bandit uncertainty.
>
> That said, it is unclear to us how mild additional structural assumptions could be that would enable $\sqrt{T}$ regret bounds.    At least for some problems, we expect semi-bandit feedback would be sufficient to design algorithms with $\sqrt{T}$ regret bounds. It would depend on specific applications as to whether semi-bandit feedback would be available or not.
>
> > The resilience definition appears strong/uniform (it is stated for all subsets), while the online reduction only needs accuracy on the queried subsets; this mismatch could confuse readers and should be reconciled more explicitly.
>
>
> **Response.**
> We appreciate the reviewer highlighting the apparent mismatch between the uniform resilience definition and the fact that the online reduction only ever queries a finite set of actions. This is intentional since the goal of the resilience definition is to serve as a purely offline, black-box abstraction that is independent of how the algorithm is later deployed online. This allows the offline algorithm to be reused without exposing or constraining its internal query pattern.
>
> > The algorithm requires (or implicitly uses) knowledge/upper bounds of $\delta$, N, h (and horizon T) to set exploration length; it is unclear how robust the method is to misspecification or how to tune in practice.
>
> **Response.**
> We agree that the algorithm assumes knowledge (or upper bounds) on these parameters in order to set the exploration length $m$. This is standard in oracle-based offline-to-online reductions and explore-then-commit bandit methods, while note that an upper bound on these values can be used. Regarding the specific parameters mentioned,
> - $\delta$ is based on the reward/constraint classes, action space,  and the offline approximation algorithm used; the reward/constraint classes and the action space are widely assumed to be known by learner in the MAB literature, the latter is chosen by the learner.
> - $N$ is based on the offline approximation algorithm used, which is chosen by the the learner, and the action space which is typically assumed known.
> - $h$ is based on the reward/constraint function classes, which are widely assumed to be known by the learner.
> - $T$ is the horizon and is also commonly assumed to be known in the MAB literature, though a number of methods (for simpler problems) do have any-time guarantees.  There is a common doubling-epoch meta-algorithm that converts horizon-dependent methods into any-time methods.
>
>
>
> > The paper is heavily theoretical and (as provided) does not include empirical validation; while not required, experiments could materially improve confidence and interpretability of constants and regimes.
>
> **Response.**
> We appreciate the suggestion. Given the theoretical focus of the paper, we have prioritized formal guarantees and proof techniques.

---

> > ### Author Response · Authors · 2026-02-19
> >
> > > Requested Change 1: Clarify the stochastic model and independence assumptions used in Theorem 3 (and related results).
> >
> > **Response.**
> > We have revised the sentence in the stochastic model to clarify this. The revised sentence is:
> >
> > ``Let $\Omega$ be a ground set of $n$ base arms and at each time step $t$, the learner selects an action $A_t \subseteq \Omega$ and observes a stochastic reward $f_t(A_t) \in [0, f_{\max}]$ and a cost $ g_t(A_t) \in [0, g_{\max}] $, both drawn from unknown distributions (assumed independent across time for each fixed action) with expectations $f(A) = \mathbb{E}[f_t(A)]$ and $g(A) = \mathbb{E}[g_t(A)]$."
> >
> > We do not need i.i.d. across time, just independence. Further, independence between functions $f_t$ and $g_t$ is not needed in the analysis.
> >
> >
> > > Requested Change 2: Reconcile the resilience definition (uniform oracle error for all sets) with what the online reduction actually requires (accuracy only on queried sets).
> >
> > **Response.**
> > We have added a remark to address this. The added remark is:
> >
> > ``Definition 1  requires the approximation guarantees to hold uniformly over all subsets \(S \subseteq \Omega\), even though the online reduction in Section~5 only ever queries the offline algorithm on a finite (data-dependent) collection of sets. This stronger, uniform formulation is intentional. It allows resilience to serve as a purely offline, black-box abstraction that is independent of the internal query pattern or adaptivity of the algorithm when deployed online. In particular, the online reduction only uses the resilience guarantees on the sets actually queried, but uniform resilience ensures that these guarantees hold without requiring the online learner to reason about or restrict the offline algorithm’s behavior. We note that weaker notions of resilience restricted to queried sets would suffice for a fixed algorithmic instantiation, but would entangle the offline property with the online execution and reduce composability."
> >
> >
> > We also believe that assessing the robustness of (offline) approximation algorithms is of independent interest.  When such algorithms are used in practice, the assumptions of exact oracles may be unrealistic in some settings (such as if values are estimated from data).  A decision maker might chose to use an approximation algorithm A that is robust but either slightly slower or has a slightly worse approximation bound than another approximation algorithm B if B is highly sensitive to oracle errors.

---

### Review · Reviewer_wgch · 2026-02-17

**Summary Of Contributions:**

## **Summary**
This paper examines bi-criteria combinatorial decision making in the bandit setting. At each step, a learner chooses a combinatorial action  and receives only the sum of two values: an objective and a constraint-related metric , both of which are noisy. The learner aims to achieve close to the optimal offline solution for the objective while maintaining a small total constraint violation. The key abstraction is the general concept of resilience for offline bi-criteria approximation algorithms.
An offline bi-criteria approximation algorithm is resilient if it still produces approximately optimal and approximately feasible solutions even if it has slightly imprecise oracle access to the objective and constraint functions. This definition provides a clean interface: once an offline algorithm is shown to be resilient, it can be easily transferred to the online bandit setting.
Based on this insight, this paper provides a black-box offline-to-online transfer. The above reduction implements the offline algorithm by replying to each of its oracle queries with empirical averages computed by repeatedly executing the queried action. After a controlled number of rounds on this exploration step, the learner locks in the set returned by the offline algorithm
for the remaining rounds. The analysis shows that the resulting online procedure achieves sublinear objective regret and sublinear cumulative constraint violation, with rates that depend on (i) the offline algorithm's query complexity and (ii) its resilience parameters. Finally, the framework is instantiated for several important submodular and fairness-constrained problem classes by proving that standard greedy-style offline algorithms satisfy the required resilience conditions. Overall, the paper offers a general and modular path from resilient offline
bi-criteria approximation to online bandit algorithms with provable guarantees.

## **Key Comments**
This paper provides a strong theoretical contribution with a clean and well-motivated message: formalize a robust offline concept of resilience and then leverage it to build an online solution for bi-criteria combinatorial bandits under bandit feedback. The message is clean, and the problem-specific instantiations are believable, and the authors pay close attention to the technical details, including identifying and correcting a commonly repeated but incorrect generalization in the literature.

The limitations are mostly quantitative and interpretational. On the quantitative side, the results may contain large constants due to the dependence on the offline query complexity $N$  and the resilience parameter $\delta$. On the interpretational side, the paper’s “adversarial” terminology for oracle noise may raise the bar for what the online theory actually assumes, since the reduction ultimately uses bounded i.i.d. stochastic feedback to achieve concentration during exploration.

**Additional Comments:**

N/A

**Audience:**

Yes

**Audience Explanation:**

The paper is timely and offers a solid mathematical framework for the problem. I am confident the community will benefit from these results.

**Claims And Evidence:**

Yes

**Claims Explanation:**

I have briefly reviewed the proofs, and to the best of my understanding, they seem correct.

**Requested Changes:**

In the paper, “adversarial” mainly refers to worst-case, bounded oracle inaccuracies in the offline resilience definition i.e., the offline algorithm remains approximately correct even if the oracle values are perturbed within a small neighborhood. But the online reduction and regret/CCV guarantees rely on stochastic i.i.d., bounded feedback so that repeated plays yield the desired concentration bound. This, in my opinion is a stochastic bandit assumption, not an adversarial bandit model. So a reader may interpret “adversarial” as meaning adaptive, non-stochastic bandit noise, and expect techniques/guarantees that hold under that stronger setting. That mismatch can cause confusion about what is actually being proved.

---

> ### Author Response · Authors · 2026-02-19
>
> Thank you for your time and feedback.  Below we reply to your requested change.
>
>
> > In the paper, “adversarial” mainly refers to worst-case, bounded oracle inaccuracies in the offline resilience definition i.e., the offline algorithm remains approximately correct even if the oracle values are perturbed within a small neighborhood. But the online reduction and regret/CCV guarantees rely on stochastic i.i.d., bounded feedback so that repeated plays yield the desired concentration bound. This, in my opinion is a stochastic bandit assumption, not an adversarial bandit model. So a reader may interpret “adversarial” as meaning adaptive, non-stochastic bandit noise, and expect techniques/guarantees that hold under that stronger setting. That mismatch can cause confusion about what is actually being proved.
>
>
>
> **Response.**
> Thank you for pointing this out. We agree that our use of the phrase "adversarial" for the noise in the offline resilience analysis could be confusing to readers.  We  replaced that phrase in the paper, mentioning possibly worst-case bounded noise.
>
>
> The term adversarial in the paper refered exclusively to the offline setting, namely, to worst-case, bounded perturbations of the objective and constraint oracles in the definition of resilience. In this sense, resilience is a stability property: the offline algorithm must remain approximately optimal and feasible even when oracle values are perturbed in an arbitrary (non-stochastic) manner within a prescribed error tolerance.
>
> In contrast, the online learning model and regret/CCV guarantees are fully stochastic, relying on bounded, independent feedback so that empirical averages concentrate around their expectations. The online reduction does not claim robustness to adversarial or adaptive bandit feedback.
>
> We have worked on clarifying this explicitly. In the Introduction, we added: ``The notion of resilience captures robustness of offline bi-criteria approximation algorithms to worst-case, bounded oracle perturbations, while the online reduction operates in a stochastic bandit setting where repeated plays enable concentration."
>
> Further, we added explicitly in Section 5 that ``For the offline resilience definition (Definition 1), we assume bounded oracle perturbations. In contrast, the online  model studied in this section assumes stochastic bandit feedback: for each fixed action, repeated plays yield independent, bounded observations with fixed expectations, enabling concentration of empirical means. The regret and cumulative constraint violation guarantees in this section rely on this stochastic feedback model and do not extend to adversarial or adaptive bandit noise."

---

### Review · Reviewer_ennZ · 2026-02-27

**Summary Of Contributions:**

The authors study study bi-criteria combinatorial optimization under noisy function evaluations introducing a new notion of resilience that captures the complexity of the problem. They achieve regret of order $O(\delta^{2/3} N^{1/3} T^{2/3})$. (Furthermore they establish  resilience guarantees of several greedy algorithms).

**Additional Comments:**

I think is an excellent paper that provide a theoretical contribution for the field.

**Audience:**

Yes

**Audience Explanation:**

Online learning (and bandits as a part of it) is a field that is expanding and stands as cutting edge research (see the works of Lattimore, Hazan, also  Bach's for submodular optimiztion) in machine learning research

**Broader Impact Concerns:**

There are no broader impact concern

**Claims And Evidence:**

Yes

**Claims Explanation:**

The proofs seem convincing in line with the claims. There are no overstatements.

**Requested Changes:**

The paper is clear and properly justified, The scopes are well and explained and also the contributions are stated clearly without overstatements. Many concepts seems genuinely new and well investigated and it seems to me overall a significant theoretical contribution to the literature, I have appreciated how the resilience definition enables an offline-online conversion of the analysis. Since the regret are new there' no state of the art to compare it, but from what I now from similar literature are in line with the lower bounds of similar setting. The notation is clear (and adopts most of the standard from bandit and online learning literature) and the related works are properly discussed. While I understand the focus is mainly theoretical I would have liked to see some experiments, even in the most speculative works on bandits (see some Kaufmann works) some empirical validation is provided. In order to make the exposition cleaner I would suggest to condense the proof of Theorem 3, and leave mainly hints and highlights of the parts of the proof you think may be interesting outside the scope of the work itself. As I said I think the contributions are clear, but it would be even better if the new analysis techniques they've introduced were discussed inthe broader framework of the literature, explaining the progress with respect of previous literature that relies on exact submodularity (in order to better explain myself, is often the case that in bandit learning some new concentration inequality is derived, the authors usually higlhlights this new contribution and explain how can be used outside the scope of the work itself).

---

> ### Author Response · Authors · 2026-03-05
>
> We are grateful for your time and feedback.  Below we reply to the requested changes.
>
> > The paper is clear and properly justified, The scopes are well and explained and also the contributions are stated clearly without overstatements. Many concepts seems genuinely new and well investigated and it seems to me overall a significant theoretical contribution to the literature, I have appreciated how the resilience definition enables an offline-online conversion of the analysis. Since the regret are new there' no state of the art to compare it, but from what I now from similar literature are in line with the lower bounds of similar setting. The notation is clear (and adopts most of the standard from bandit and online learning literature) and the related works are properly discussed.
>
> **Response**
> Thank you, we are glad to hear you appreciated our work.
>
> > While I understand the focus is mainly theoretical I would have liked to see some experiments, even in the most speculative works on bandits (see some Kaufmann works) some empirical validation is provided.
>
> **Response** We acknowledge  that even for works focusing on theoretical contributions, empirical results can be beneficial to readers as a lens to view improvements.  In addition to the theoretical focus of the paper, since our algorithm for the online setting is a meta-algorithm and there is no prior work (namely no baselines with non-trivial regret and constraint violation guarantees for the example problems), we focused on formal guarantees and proof techniques.
>
>
> > In order to make the exposition cleaner I would suggest to condense the proof of Theorem 3, and leave mainly hints and highlights of the parts of the proof you think may be interesting outside the scope of the work itself.
>
> **Response**
> We have condensed the proof for Theorem 3 in the main paper to improve exposition, leaving the full proof in the appendix.
>
>
> >  As I said I think the contributions are clear, but it would be even better if the new analysis techniques they've introduced were discussed inthe broader framework of the literature, explaining the progress with respect of previous literature that relies on exact submodularity (in order to better explain myself, is often the case that in bandit learning some new concentration inequality is derived, the authors usually higlhlights this new contribution and explain how can be used outside the scope of the work itself).
>
> **Response.**
>
> We appreciate this suggestion.  We do believe the techniques introduced in this work can inspire resilience analyses for approximation algorithms designed for related problems.  However, it is not clear to us if they would be beneficial to the broader literature on submodularity as stand-alone results in the same way that a new concentration inequality could be.  The offline-to-online framework itself is valuable, as researchers studying (offline) submodular optimization can design a (bi-criteria) approximation algorithm for their problem and, if it is shown to be resilient, the cumulative regret and constraint violations guarantees for the online problem will follow immediately from Theorem 3.

---

### Decision · Action_Editor_o4KK · 2026-04-02

**Recommendation:** Accept as is

**Additional Comments:**

This paper studies bi-criteria combinatorial optimization with bandit feedback, where the technical challenge originates from the coupling of the two problem structures: the degradation of objective and constraint guarantees are coupled under noise. The key idea of this work is a new formulation of a resilience concept for bi-criteria approximation algorithms, and based on that a general offline-to-online conversion framework is provided, achieving both sublinear regret and sublinear cumulative constraint violation. This general pipeline is finally instantiated on several iconic special cases.

All reviewers found this paper correct and clear, and the claimed contributions are valid. These meet the TMLR acceptance criteria. As the result, a consensus has been reached to accept this paper.

There are however several noted weaknesses which the authors may consider addressing in the revised version or future works.

- It is noted that the defined resilience is likely stronger than what is actually needed in the subsequent analysis. For example the former is based on non-stochastic characterizations of the noise whereas the latter assumes iid. Therefore optimality is not analyzed in the present work, and the presence of $\delta$ and $N$ in the final result makes it less interpretable.

- Although the obtained quantitative results are new, it is noted that the key scientific message and methodological novelties are not clear enough. In some sense the approach of this paper is not particularly surprising although it is technically solid.

- The reviewers recognized that the main contributions of this paper are theoretical, but still adding experiments would be appreciated.

**Audience:**

Yes

**Audience Explanation:**

All reviewers found the topic of the paper relevant to the theoretical side of the TMLR community and the contributions of the paper valid.

**Claims And Evidence:**

Yes

**Claims Explanation:**

All reviewers found the paper clearly written and supported by correct and convincing evidence.